# SIMPLICITY is an agent-based, multi-scale mathematical model to study SARS-CoV-2 intra- and between-host evolution
Pietro Gerletti [1,2] ✉, Nils Gubela [2,3,5], Jean-Baptiste Escudié[1,5], Denise Kühnert[1,6] & Max Von Kleist [2,4,6]

Computational tools are frequently used to describe pathogen evolutionary dynamics either within infected hosts or at the population level. However, there is a lack of models that capture the complex interplay between within-host and between-host evolutionary dynamics, leaving a knowledge gap with regard to realistic evolutionary dynamics. We present SIMPLICITY, a multi-scale mathematical model that combines within-host disease progression and viral evolution with a population-level model of virus transmission and immune evasion. We parameterize SIMPLICITY based on SARS-CoV-2 within-host viral dynamics, observed evolutionary rates, and dynamics of immune waning. We then apply it to study the dynamics and mechanisms driving SARS-CoV-2 evolution at the population level. We compare a baseline toy model of gradually increasing transmission fitness with an adaptive fitness landscape model that accounts for infection history and immune waning. Our simulations demonstrate that escape from population immunity generates evolutionary dynamics encompassing selective sweeps, which resemble SARS-CoV-2 evolution.

During the last decades, mathematical modeling of infectious diseases has become an important tool in public health for understanding the dynamics of disease transmission, for assessing the impact of interventions, and to propose public health policies[1,2]. During the SARS-CoV-2 pandemic, the importance of mathematical models that can describe intra-host viral dynamics, as well as transmission patterns and evolutionary dynamics, became ever more evident. Such models were used to predict transmission dynamics[1,3], develop containment strategies[4], help forecasting case-numbers and vaccine efficacy[5,6], and quantify dominant selection forces[7], to name a few.

Mathematical modeling of infectious diseases can take many forms, depending on their scope, granularity and scale[1,2,8]. Susceptible-Infected-Recovered (SIR) models divide the population into compartments and describe transitions between compartments through differential equations[9]. They are commonly used to model outbreaks that can be described by assuming homogeneous contact networks. These models can be extended to more complex scenarios, for example, by accounting for acquired immunity through infection or vaccination, or by considering more complex contact networks, as well as stochastic dynamics[10]. For example, in 2020, Chinazzi et al. used a disease transmission model to understand the impact of travel restrictions on the spread of SARS-CoV-2 in China and globally[5]. Moore et al. used a compartmental, age-structured model to model optimal vaccination strategy in the UK[6]. At the finest granularity, agent-based models are used to simulate individual-level interactions within a population, giving more insight into how mobility or individual behavior affects the dynamics of an outbreak[11,12].

Mathematical modeling can also be used to study within-host pathogen dynamics[4,13], i.e., how the pathogen responds to antiviral drugs[14–16], interacts with the immune system, or for investigating the pathogen's within-host evolution dynamics[17,18]. For example, van der Toorn et al. developed a statistical model of SARS-CoV-2 intra-host viral dynamics to help policymakers develop recommendations for non-pharmaceutical interventions at the individual level[4].

Lastly, a number of approaches aim at predicting population-level evolutionary dynamics by combining compartment, or agent-based models, with simple models of viral evolution[8,19,20]. However, these models rarely take within-host viral dynamics into account. Notably, within-host viral dynamics may have profound implications for the population-level evolution: for SARS-CoV-2, we observed near-identical viral genomes in typical outbreak scenarios[21], which may explain gradual- or slow population-level evolution over extended periods of time[22]. However, many variants of concern, which are highly divergent from circulating lineages at their time of

[1]Center for Artificial Intelligence in Public Health, Robert Koch Institute, Berlin, Germany. [2]Department of Mathematics & Computer Science, Freie Universität Berlin, Berlin, Germany. [3]International Max-Planck Research School "Biology and Computation" (IMPRS-BAC), Max-Planck Institute for Molecular Genetics, Berlin, Germany. [4]Project groups, Robert-Koch-Institute, Berlin, Germany. [5]These authors contributed equally: Nils Gubela, Jean-Baptiste Escudié. [6]These authors jointly supervised this work: Denise Kühnert, Max Von Kleist. ✉e-mail: simplicity.twisty120@passfwd.com

emergence, are believed to have arisen in chronically infected, immune-suppressed individuals[23]. Such variants may subsequently cause selective sweeps at the population level by circumventing prevalent immunity at the time of their emergence[7]. Aside from SARS-CoV-2, intra-host dynamics are important for other viral pathogens, such as HIV, as well as some bacterial infections[24,25]. These observations highlight a need to better understand the impact of within-host processes on between-host evolutionary dynamics.

Phylodynamic approaches address the inverse problem by learning evolutionary models from molecular surveillance data[26]. These approaches, ranging from genomic epidemiology to phylogenetics, provide us with a quantitative description of population-level evolutionary parameters, averaging over potentially heterogeneous within-host dynamics. While these approaches generate meaningful interpretations of actual outbreaks or evolutionary trajectories, there is a lack of ground-truth data to challenge interpretations derived from phylodynamics approaches or to define their scope and limitations.

Multi-scale models that can capture the relationship between within-host and between-host viral dynamics may provide a tool to generate ground truth data, while delivering mechanistic insight into evolutionary dynamics[27–30]. Having a framework to simulate the virus spread in a population coupled to its evolutionary dynamics both within and between hosts can be a powerful tool to study the intricate mechanisms that couple virus evolution and transmission.

Here, we present an agent-based model of population-level and intra-host processes, enabling us to generate realistic simulations of virus evolution. The model, StochastIc siMulation of sars-cov-2 sPreading and evoLutIon aCcountIng for wiThin-host dYnamics (SIMPLICITY), was developed for SARS-CoV-2 and includes four components: (i) an agent-based SIRD model that simulates the spread of the virus at the population level, (ii) a stochastic intra-host model of disease progression within individuals, (iii) an evolutionary model of SARS-CoV-2, which introduces mutations into the within-host dominant lineages, and (iv) a phenotype model that relates the viral genome to a transmission fitness. By integrating these components into a single model, we aim to gain a deeper understanding of the mechanisms driving SARS-CoV-2 evolution, which could lead to more effective public health interventions and future pandemic response strategies. At the same time, we want to provide the modeling framework for other scientists to be able to run their own simulations on SARS-CoV-2 epidemics, to produce synthetic data, and to adapt the model to other viruses. In this paper, we present the theoretical framework and provide SIMPLICITY as a software package specifically within the context of SARS-CoV-2.

## Results
### Overview

In this paper, we present SIMPLICITY, a multi-scale mathematical model combining models of intra- and between-host virus evolution and population infection spreading dynamics. Figure 1 highlights the four core parts of SIMPLICITY: the population modeling layer of SIMPLICITY is a SIRD compartment model, which simulates the population dynamics of a viral outbreak in a human population. Each individual who enters the infected compartment will progress in the clinical stages described by the intra-host model taken from the work of Van der Toorn et al.[4] (Fig. 2A). We model evolutionary processes using a nucleotide substitution model that can be fine-tuned to investigate different viral evolution scenarios. The genotype of simulated lineages that evolve during a SIMPLICITY simulation influences the spreading dynamics of the virus through a relative transmission fitness score, calculated with the phenotype model. We implemented two different phenotype models: a baseline phenotype model and an immune-waning phenotype model (see Methods). We set up and run two experiments: the first one with the goal of fine-tuning the model parameter nucleotide substitution rate (NSR) to reproduce the observed nucleotide substitution rate (OSR) of the SARS-CoV- 2 pandemic. In the second experiment, we explored the effect of population immunity landscape on SARS-CoV-2 lineage emergence and showed that immune waning dynamics generate selective sweeps.

### Reproducing Observed Substitution Rates

In the first experiment, we (i) individuate the NSR value range that we can use to reproduce real-world SARS-CoV-2 substitution rates, and (ii) developed a model fine-tuning pipeline that allows simulations to reproduce any desired OSR. Figure 2B shows how an estimate of the OSR can be obtained using a linear regression run on the simulated sequencing dataset produced by a single simulation run (OSR is the line slope) with parameters given in Supplementary Table S1. As the OSR does not exactly match the NSR due to the influence of population-level transmission events and the intra-host dynamics, we explored their relationship by estimating the OSR for a range of NSR. Based on AIC, the best fit for the NSR/OSR relationship among five different functions (linear, logarithm, exponential, tangent, and a polynomial spline) was an exponential function (Supplementary Table S2). Taking the inverse of the fitted curve, we obtained the NSR value needed to run simulations that will produce the target OSR for the given set of parameters (Fig. 2C). As can be seen in the figure, the exponential curve fits the data very well, showing that we can reproduce any OSR value,

**Fig. 1 | Overview of SIMPLICITY.** In the upper part of the figure, one can see the compartments of the SIRD model, which consist of Susceptible (white), Infected (red), Diagnosed (yellow), and Recovered (green). These compartment trajectories over time simulate the population dynamics of a SARS-CoV-2 viral outbreak in a human population. Once an individual enters the infected compartment, they will progress through the intra-host model that simulates their clinical dynamics. In the intra-host model, groups of compartments represent a specific clinical state (pre-infectious/infectious/post-infectious) that an individual infected with SARS-CoV-2 will go through. In a SIMPLICITY simulation, we model virus evolution (bottom right). The virus genome (or selected genes) evolves over time as it spreads between agents. The phenotypic model (bottom left) connects the virus genomes to the infection spreading dynamics by assigning a relative transmission fitness score to each lineage, based on their genotype. Graphic elements: Pixabay (coronavirus icon), Diemen Design (human icon), Veesler et al.[48] (spike protein structure).

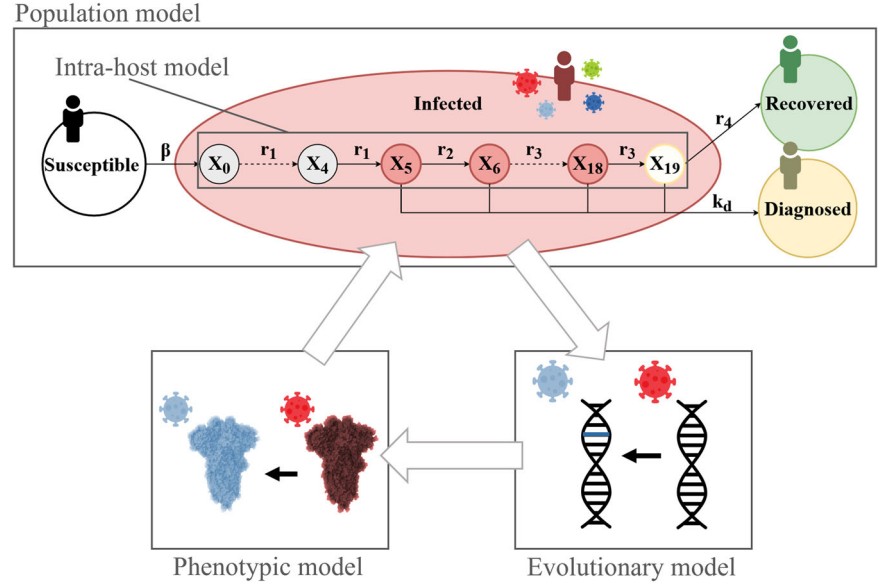

**Fig. 2 | Model parametrization. A** Solution of the intra-host model, parametrized on German clinical data. Each curve shows the probability of an individual being infectious, detectable, or recovered t days after the infection. **B** Linear regression between simulated sequencing data timestamps and genomic distance from the founder genome ($N = 1947$). The slope of the regression is an estimator of the per-site per-year evolutionary rate. In our model, we denote it Observed Substitution Rate (OSR). We use this regression method to find OSR point estimates for each simulation run and then fit the value of the model Nucleotide Substitution Rate (NSR) to the OSR. **C** NSR/OSR exponential regressor fit (log-log space). Each point in orange is a single OSR estimate from a simulation run with a specific NSR (x-axis). We used 15 different NSR values, each simulated N=50 times. The exponential curve fits the average OSR for each value of the NSR and the OSR estimate obtained by applying the regression shown above to joint simulation data. With this regressor, the user can select an NSR that corresponds to an OSR between $10^{-4}$ and $10^{-3}$ (shaded in gray, SARS-CoV-2 substitution rate range in real data) for the given parameters set. In red we show the value used for the experiment presented in this paper.

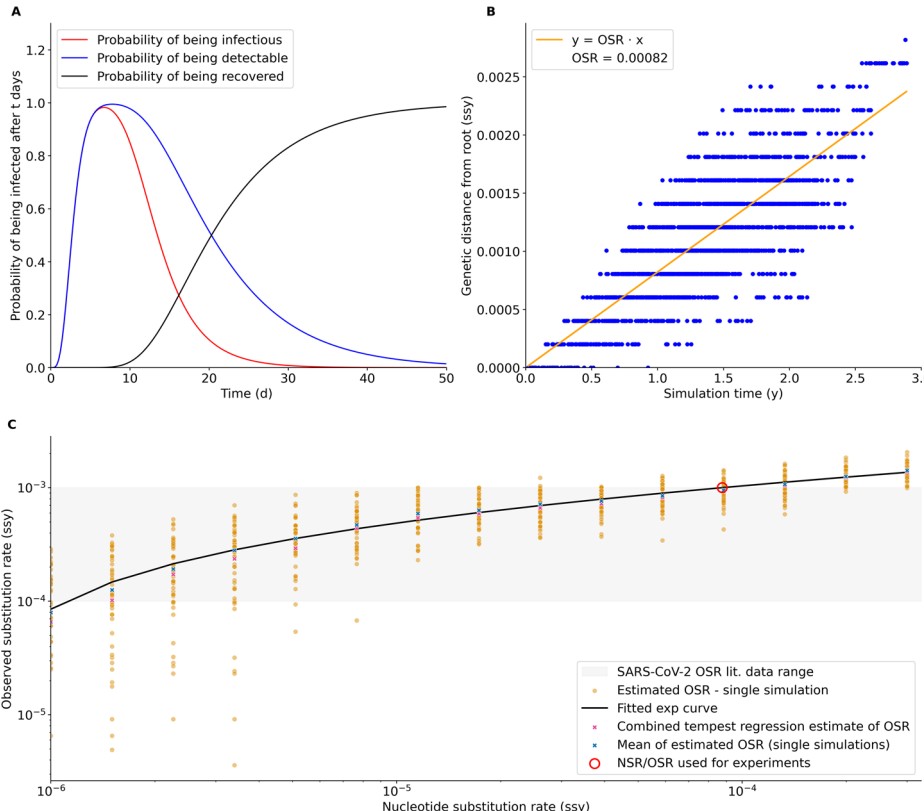

including the ones that were observed during the SARS-CoV-2 pandemic. We then fixed the NSR to 0.00008759 (OSR = $1 \times 10^{-3}$])[22] to run the second experiment, with the goal of investigating the importance of population immunity landscape in the evolution of new lineages with high spreading potential.

## Immune waning dynamics generate selective sweeps

To assess how selection may shape the evolutionary dynamics of SARS-CoV-2, we simulated SARS-CoV-2 outbreaks using two phenotype models (see Methods). In simulations with the baseline phenotype model, the virus gained fitness as it diverged away from the founder ('wild-type').

Interestingly, this tended to produce evolutionary dynamics in which no single virus lineage dominated, but rather the population of circulating lineages diversified (Fig. 3A). This tendency is also reflected in Fig. 4A, which shows the trajectory of the system entropy (calculated using lineage frequency) over a single simulation run. In this simulation scenario, the entropy increases steadily as the evolutionary landscape becomes more fragmented. Clustering single substitution lineages into 'super-lineages' (i.e., lineages that share at least 5 substitutions), did not affect this tendency (Fig. S3).

We then ran SIMPLICITY with the 'immune waning model', where a lineage had a transmission advantage if it was distinct from the lineages that appeared within the previous months (weighed by immune waning pharmacokinetics). Interestingly, considering immune waning dynamics produced selective sweeps, in which dominating lineages are replaced by waves of lineages that escape predominant immunity (Fig. 3C). Figure 4C shows how the entropy trajectory in these simulations tends to swing in waves, instead of continuously increasing. In this case, lineage clustering becomes more pronounced: minority lineages are more frequently grouped together, as they stem from the same phylogenetic tree branch as the dominant lineage driving the current wave (Fig. S3). This implies that the immune-waning model changes the evolutionary trajectory from a simple drift from the founder sequence to a more selective, directed evolutionary pathway.

Figure 3 shows a representative example (random seed = 7) of a simulation outcome using the baseline phenotype model (upper panel) and the immune-waning model (lower panel). While the trajectories of the infected compartments (A, C subplot 1) are similar between models, marked differences emerge in the average fitness score (A, C subplot 2), lineage frequency subplots (A, C subplot 3) and lineage-specific $R$ effective (A, C subplot 4). In the baseline phenotype model, the average fitness score increases gradually, plateaus, and then increases again. Over time, the baseline model produces an increasingly diverse evolutionary landscape, with no single lineage infecting more than 30% of the population by the end of the simulation. In contrast, the immune-waning phenotype model shows markedly different evolutionary dynamics. After an initial linear increase in average fitness, when population immunity is still negligible, the system enters a wave-like phase driven by immune escape. New dominant lineages periodically replace earlier ones (selective sweeps) as population immunity builds and wanes. We quantified the number of selective sweeps happening in each simulation batch ($n = 206$), after filtering out simulations shorter than 300 days, and plotted the two groups' distribution as violin plots (Fig. 4B). We used a Mann-Whitney two-sided statistical test to ensure that the differences were statistically significant, and obtained a p-value of $10^{-4}$. Subplots 4 (A, C) show the $R$ effective value for the population (average) and for each lineage over the course of the simulations. In the baseline phenotype model simulation, one can observe many frequent $R_{lineage}$ peaks, which correspond to new lineage emergence events; in comparison, the immune-waning model displays fewer but more pronounced spikes in $R_{lineage}$, aligning with the wave-like immune-escape dynamics.

Finally, the phylogenetic tree of the simulations, shown in Fig. 3A, B gives a complete overview of the evolutionary trajectory of simulated lineages. The baseline model tree shows a more scattered time of emergence (indicated by coloring), implying that temporally concurrent lineages populate distant branches on the phylogenetic tree. The immune waning model tree instead shows more visually homogeneous color clusters,

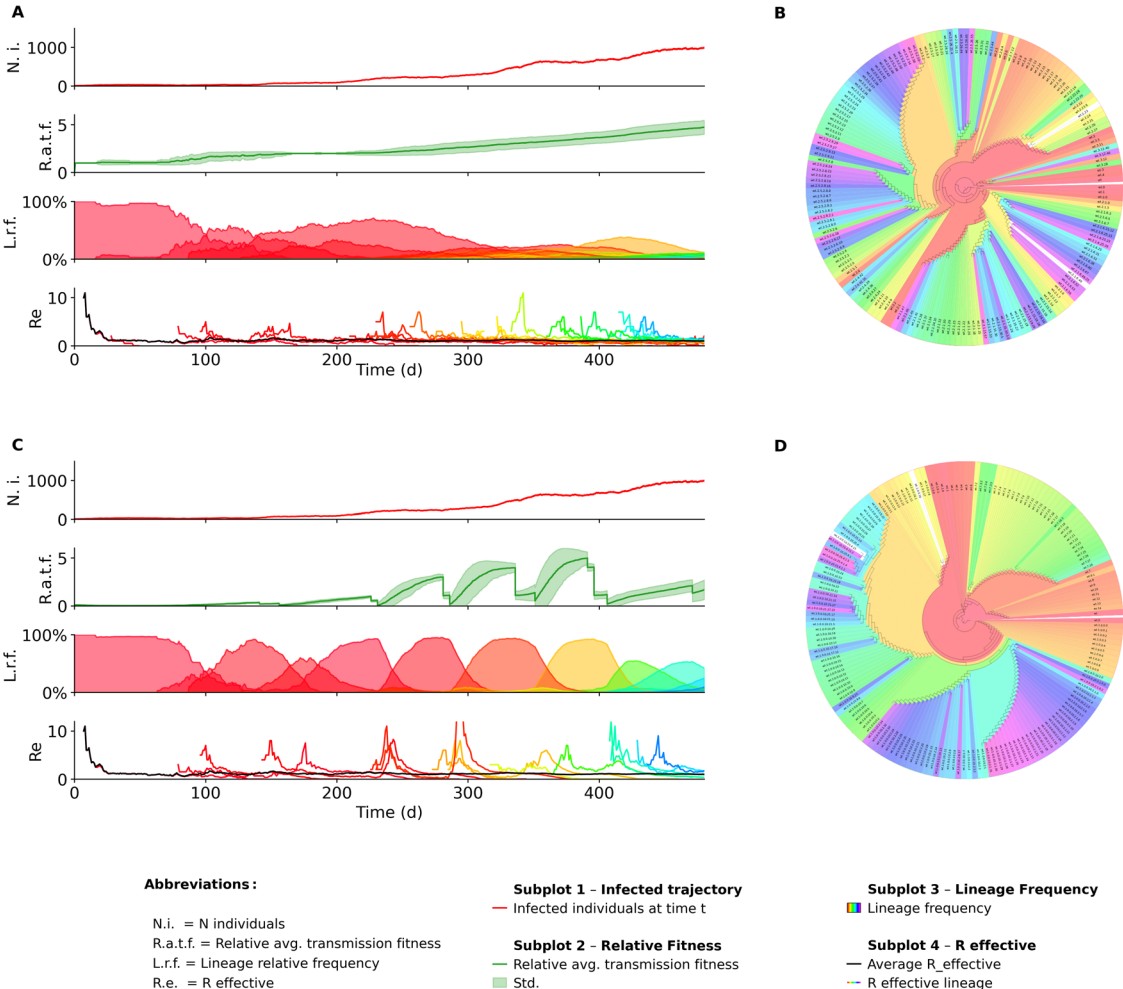

**Fig. 3 | Single simulation of a SARS-CoV-2 outbreak using different phenotype models.** **A** Baseline phenotype model results: Dynamics of infected compartments showing similarity to the immune-waning model. Average fitness, with substantial differences compared to the immune-waning model. Lineage frequencies showed a diverse viral population, with no single lineage exceeding 30% prevalence by the end. Lineage-specific effective reproduction numbers, highlighting variations between models. **B** Phylogenetic tree from the baseline model, exhibiting dispersed color patterns (time of emergence), suggesting lineages emerging at the same time are distributed across distant branches. **C** Immune-waning phenotype model results: Similar infected compartment dynamics to the baseline model. Average fitness variations from the baseline model. Lineage frequencies, illustrating how immune waning leads to selective sweeps. Lineage-specific effective reproduction numbers, showing significant model-dependent differences. **D** Phylogenetic tree from the immune-waning model, displaying cohesive color clusters, indicating that contemporaneous lineages are more closely related and tend to originate from the same branch.

indicating how lineages circulating at the same time tend to belong to the same branch of the tree and are thus more closely related.

## Discussion

Our model (SIMPLICITY) combines classical epidemiological models, intra-host disease dynamic modeling, and models of viral evolution into a cohesive framework that we parameterized for SARS-CoV-2. The experimental results demonstrate that SIMPLICITY captures the dynamics of SARS-CoV-2 at multiple biological scales, such as intra-host clinical states, population spreading, and evolutionary events (selective sweeps). SIM-PLICITY employs the Extrande algorithm for the exact simulation of inhomogeneous Poisson processes, while saving compute time in comparison to state-of-the-art methods (i.e., Gillespie's algorithm). The framework is memory-efficient and runs on standard hardware. Validation tests confirmed the mathematical robustness and consistency of SIMPLICITY. The adapted intra-host model reproduced the same average infection and infectious periods reported in the original publication[4], indicating that the modifications preserved the host disease dynamics. As expected, individuals in the diagnosed compartment exhibited shorter residence times in the infected phase, consistent with model assumptions of detection and isolation. At the population level, the effective reproduction number $R_{effective}$ matched the $R$ parameter value specified for the simulations (within 10% variability), confirming that the model behaves as intended.

Users may utilize accessible high-level parameters like the reproduction number $R$ to adjust internal model parameters (in particular, the infection rate). This allows to easily setup and customize simulation scenarios. Likewise, we provide a method to fine-tune the model to any desired evolutionary rate. While simple, the root-to-tip regression performed on the simulation time-stamped sequencing data to estimate the OSR works well and allows SIMPLICITY simulations to be tuned to accessible, real-world pandemic data. SIMPLICITY provides both infection and phylogenetic trees (Fig. S2).

Agent-based models such as nosoi[31] simulate transmission chains, but do not model viral evolution or intra-host dynamics. Tools like VGsim[32] and FAVITES[33] incorporate viral sequence evolution and can generate phylogenetic trees, but they either simplify within-host processes or treat them independently from epidemiological dynamics. The Opqua framework[34] models genome-level viral evolution with intra-host selection, but it does not include a compartmental between-host model, and simulations with more than $10^3$ individuals rapidly become computationally intractable,

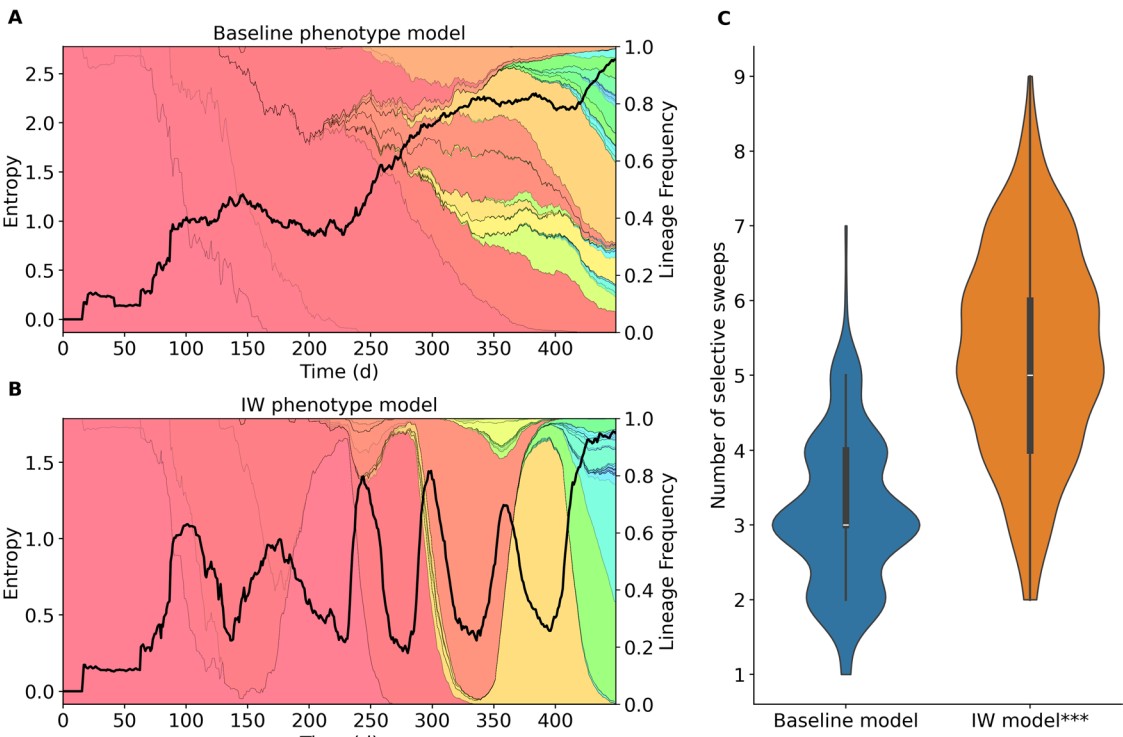

**Fig. 4 | System entropy and selective sweeps violin plot. A** Entropy trajectory from a single simulation (random seed = 7) run under the baseline model, plotted over a stackplot of all lineages frequency. Entropy, calculated from lineage frequencies, increases steadily over time, reflecting a progressively fragmented evolutionary landscape. **B** Simulations under the immune-waning model display wave-like fluctuations in entropy, corresponding to cycles of lineage dominance and replacement (selective sweeps). **C** Violin plots comparing the distribution of selective sweeps count between the two models across multiple simulations ($N = 206$). Boxplots encompass the inter-quartile ranges (IQR), central dots depict the median, and whiskers extend to 1.5 times the IQR. A two-sided Mann-Whitney test confirmed that the difference between the two groups is statistically significant ($p = 2.62 \cdot 10^{-4}$).

while SIMPLICITY can handle simulations that are two orders of magnitude larger. Another model, e3SIM, integrates epidemiological, ecological, and evolutionary processes in a scalable agent-based framework with explicit contact networks, while using discrete-time forward simulation (which may result in numerical errors)[35]. SIMPLICITY bridges the gap between these approaches, with a focus on SARS-CoV-2, a virus that has impacted public health in unprecedented ways: it integrates an SIRD model, a within-host clinical dynamics model, and mutation-driven evolutionary processes within a stochastic, agent-based framework. This allows thorough investigation of how different mechanisms shape the spread of infection, transmission lineage dynamics, and evolutionary dynamics. Beyond its core functionality, SIMPLICITY enables investigation of public health interventions. By incorporating diagnosis and isolation dynamics into the intra-host model, the model reproduces reduced infectious periods for diagnosed (and isolated) individuals. This feature can be used to explore how different levels of detection impact both outbreak trajectory and evolutionary paths for the virus. This provides a tool for exploring outbreak scenarios and public health strategies, such as identifying detection thresholds that may slow or halt transmission or result in reduced lineage emergence.

Experiments using different phenotype models highlight the importance of the population immune landscape in shaping SARS-CoV-2 evolution. The baseline phenotype model produces gradually diversifying evolutionary dynamics, which may spawn into multiple evolutionary directions at the same time. Over a long time period, no single lineage dominates while Shannon entropy increases (Fig. 4). While comparing model-generated entropy predictions to real-world observational data would be valuable, the currently utilized evolutionary model may be too simplistic, and many factors, including the impact of long-shedders on viral evolution, remain to be included in SIMPLICITY. The immune-waning model assumes that immunity from past infections protects from re-infection by similar variants until immunity wanes off. However, immunity

against a dissimilar variant is waning much faster[7], such that re-infection can happen after a shorter time since the last infection. In contrast to the baseline model, we observed that the immune-waning model induces oscillatory lineage dynamics with periodic hard selective sweeps, driven by immune escape. These dynamics more closely reflect actual SARS-CoV-2 variant dynamics and our results also align with recent findings by Raharinirina et al. (2025)[7]: currently prevalent, antibody-mediated immunity and its ability to cross-neutralize emerging variants denote the major selective force driving SARS-CoV-2 evolution. Together, these findings emphasize the need to simultaneously account for infection- and immunity dynamics in evolutionary modeling of SARS-CoV-2.

Despite its versatility, the current implementation of SIMPLICITY has a number of limitations: although exact event simulation using Extrande reduces computational overhead in comparison to other exact methods, the model currently does not scale to very large population sizes. Simulations remain tractable for populations with up to 100000 cumulative infections, but larger scenarios would benefit from abstraction strategies, such as clustering individuals into super-agents, to reduce computational load and runtime. At the population level, assuming that individuals become susceptible quickly after recovery is not biologically realistic, as both variant-specific and cross-variant immunity following infection are well documented. However, in the immune-waning phenotype model, the duration of immunity is implicitly represented through the gradual loss of immune protection at the population level. This formulation therefore accounts for transient immunity without requiring explicit tracking of individual immune states. Considering the evolutionary model, in the applications presented here, viral evolution is limited to modeling substitutions (under a strict clock model and without genome site variation) and does not include recombination. Nevertheless, the model is able to accommodate site-specific rate categories. As for the clock model, one could define groups of lineages that evolve at different, user-specified rates to simulate evolution under a

relaxed clock assumption. Appropriate model parameterization would be crucial to ensure results in line with SARS-CoV-2 empirical data. Moreover, the SIMPLICITY framework lays the groundwork for future recombination modeling by incorporating dominant lineages in the host. Finally, we need to consider the abstract nature of the fitness function used in the phenotype models. We simplified highly complex evolutionary processes, which emerge from a constellation of factors that affect viral evolution, with relatively simple conceptual models, aimed at exploring specific evolutionary scenarios. In particular, in the model presented here, we do not explicitly model intra-host competition and lineage emergence, limiting it to the emergence of random intra-host lineages on which then inter-host selective pressure acts. Our modeling choice is based on the assumption that for acute respiratory viruses like SARS-CoV-2, the selective pressures at the intra-host and inter-host are largely independent from one another[36,37]. Nevertheless, for different modeling scenarios or for using SIMPLICITY with different pathogens, it would be important to adapt the intra-host evolutionary model to better reflect the underlying biology.

Future development of SIMPLICITY may focus on extending its biological realism and range of applications. One direction would be modeling long-shedders or immunocompromised individuals, i.e. hosts with prolonged infection durations who may provide an environment that facilitates the accumulation of multiple mutations and enables large evolutionary jumps. This hypothesis, which has gained traction in the literature[23,38,39], can be tested in future simulations by adapting the intra-host dynamics to include a subpopulation with persistent infection. Another possible direction is the application of SIMPLICITY to other respiratory viruses that show similar infection dynamics to SARS-CoV-2. By refitting the model parameters, SIMPLICITY could be employed to investigate viral dynamics in e.g., influenza virus, expanding its relevance for pandemic preparedness.

In this study, we presented SIMPLICITY, an open-source Python software that implements an agent-based, multi-layer, stochastic compartment model to simulate SARS-CoV-2 spread and evolution through integration of intra-host and population dynamics with genomic evolution. The model can generate ground truth data in the form of the true infection and phylogenetic trees of a simulation, together with synthetic, time-stamped sequencing data in aligned FASTA format. These model outputs can be used to test evolutionary hypotheses on the mechanisms underlying new lineage emergence, and to potentially benchmark existing phylogenetic pipelines. We were able to reproduce qualitative characteristics of SARS-CoV-2 lineage evolution, such as the selective sweep events that we observed during the SARS-CoV-2 pandemic, and to obtain simulations with observed evolutionary rates in line with real-world data, through appropriate model parametrization. Our experimental results show the importance of population immunity as a driver of SARS-CoV-2 evolution, and we invite other modelers to consider such processes in future studies.

## Methods

We developed SIMPLICITY, which consists of an epidemiological model and an intra-host model of disease progression, combined with a within-host evolution model and a model relating transmission fitness to the viral genome of infected individuals. To enable efficient simulation of this multiscale model, we adapt a rejection-based exact stochastic simulation method[40], which operates at the time-scale of the epidemiological process (akin to the work from Gubela et al.[12]), thus avoiding numerical errors, while enabling efficient computation. Rejection-based exact stochastic simulation methods are algorithmic frameworks designed to efficiently simulate systems where stochastic events occur with dynamically changing rates. The Gillespie stochastic simulation algorithm (SSA) is the foundational exact method that ensures correct sampling of both event times and reaction channels based on fixed propensities between events. However, in systems with state-dependent propensities that vary continuously, the SSA assumption of constant propensities between events no longer holds. Rejection-based methods, such as Extrande, address this by drawing candidate event times from exponential distributions using propensity upper

bounds and applying a rejection step to maintain numerical exactness. In the following sections, we will go over the details and parametrization of each part of the SIMPLICITY model.

### Intra-host model

The intra-host model denotes a semi-mechanistic stochastic transit compartment model that was previously fitted to clinical data and reflects variability in SARS-CoV-2 infection dynamics[4]. Due to its model design, the model can be solved algebraically. In brief, the SARS-CoV-2 infection time course is represented by discrete compartments $\boldsymbol{x} = (x_0, \ldots, x_n)$ with exponentially distributed waiting times. Disease progression is partitioned into five sequential phases $j$, each consisting of several sub-compartments $(x_i, \ldots, x_{i+m_j})$ that capture variability in the duration of that phase across individuals. These phases correspond to biologically distinct stages of infection: (i) pre-detection (virus not yet detectable, non-infectious, m = 5 sub-compartments, mean duration $\tau = 2.86$ days), (ii) pre-symptomatic (virus detectable and infectious, but no symptoms, m = 1, $\tau = 3.91$ days), (iii) infectious/symptomatic (detectable, infectious, symptomatic, m = 13, $\tau = 7.5$ days), (iv) post-infectious (detectable but no longer infectious, m = 1, $\tau = 8$ days), and (v) recovered. We use a total of twenty compartments, $\boldsymbol{x} = (x_0, \ldots, x_{19})$, plus a final absorbing state that corresponds to the recovered state, to represent the infection course with sufficient granularity to reproduce the empirically observed distributions of incubation time, infectious period, and test sensitivity profiles.

The probabilistic evolution of the model can be written in matrix form, yielding the master equation:

$$\frac{d}{dt}\boldsymbol{p}_t(\boldsymbol{x}) = \boldsymbol{A} \cdot \boldsymbol{p}_t(\boldsymbol{x}),$$

where $\boldsymbol{p}_t(\boldsymbol{x})$ denotes the probability that an individual is in any infection compartment $\boldsymbol{x} = (x_0, \ldots, x_{20})^T$ at time $t$ and $\boldsymbol{A}$ is the transition rate matrix (the transpose of the generator of the underlying continuous-time Markov process).

$$\boldsymbol{A} = \begin{pmatrix} -r_1 & 0 & \cdots & & \cdots & 0 \\ r_1 & -r_1 & & & & \vdots \\ \vdots & \ddots & \ddots & & & \\ & & r_1 & -r_2 & & \\ & & & \ddots & \ddots & \\ \vdots & & & & r_3 & -r_4 & \vdots \\ 0 & \cdots & & \cdots & 0 & r_4 & 0 \end{pmatrix} \begin{matrix} x_0 \\ x_1 \\ \vdots \\ \\ \\ \vdots \\ x_{20} \end{matrix}$$

Given an initial distribution $\boldsymbol{p}_{t_0}(\boldsymbol{x})$, the analytical solution is given by

$$\boldsymbol{p}_t(\boldsymbol{x}) = \exp(A\Delta t) \cdot \boldsymbol{p}_{t_0}(\boldsymbol{x})$$

for any time $t = t_0 + \Delta t$.

In $\boldsymbol{A}$, the parameters $r_j \in \{r_1, r_2, r_3, r_4\}$ correspond to micro-state transition rates $r_j = m_j / \tau_j$, where $m_j$ denotes the number of sub-compartments in each phase $j$ and $\tau_j$ denotes the mean transition time between the five major phases of infection: from pre-detection to pre-symptomatic ($\tau_1$), pre-symptomatic to infectious ($\tau_2$), infectious to post-infectious ($\tau_3$), and post-infectious to recovery ($\tau_4$). Within each phase, sub-compartments share the same rate constant as outlined above, so that the total residence time in a phase follows a gamma distribution determined by the number of sub-compartments and the rate $r_j$ of that phase.

Biologically, each compartment $x_i$ represents a micro-state along the infection trajectory. The assignment of transmission potential and

detectability follows directly from experimental viral load data used to calibrate the model (exemplified in ref. 4). Transmission becomes possible when the simulated infection has reached the pre-symptomatic and symptomatic stages (corresponding to compartments $x_5$ through $x_{18}$) and viral RNA remains detectable until later, up to $x_{19}$. This mapping ensures that model-derived quantities such as the timing of infectiousness and PCR positivity reproduce empirical data from clinical and virological studies.

### SIRD model

The SIRD (population) model integrates the intra-host model into a population model by adding two extra propensities that regulate the rate of infection of new individuals ($a_1$) and the rate of diagnosis of infectious individuals ($a_2$). Upon a positive COVID-19 diagnosis, individuals are removed from the system for the time of infection (due to self-isolation). The propensities are formalized as follows:

$$a_1 = \beta(t) \cdot |S(t)| \cdot \sum_{i=5}^{18} |x_i(t)|, \qquad (1)$$

where $\beta(t)$ is the infection rate (related to the virus reproduction number R), $|S(t)|$ is the number of susceptible individuals at time t and $|x_i(t)|$ is the number of infected agents in compartment $x_i$. Note that we only account for agents in the infectious compartments.

Similarly we derive the diagnosis propensities

$$a_2 = k_d \cdot \sum_{i=5}^{19} |x_i(t)|, \qquad (2)$$

where $k_d$ is the rate of diagnosis.

We adjust the transition rate matrix and add the diagnosis compartment to obtain:

$$B = \begin{pmatrix} -r_1 & 0 & \cdots & & \cdots & & 0 \\ r_1 & -r_1 & & & & & \vdots \\ \vdots & \ddots & \ddots & & & & \\ & & r_1 & -(r_2+k_d) & & & \\ & & & \ddots & \ddots & & \\ \vdots & & & & r_3 & -(r_4+k_d) & \vdots \\ 0 & \cdots & & \cdots & 0 & r_4 & 0 \\ 0 & \cdots & & k_d & \cdots & k_d & 0 \end{pmatrix} \begin{matrix} x_0 \\ x_1 \\ \vdots \\ \\ \\ \vdots \\ x_{20} \\ D \end{matrix}$$

After an individual recovers, a new susceptible is introduced in the population pool, keeping the sum of infected + susceptible constant during a simulation (reinfection).

### Evolutionary model

In SIMPLICITY, we only consider substitution events within-host lineages that are relevant to transmission (i.e., we do not consider entire quasi-species[41]). Herein, we only modeled the evolution of the SARS-CoV-2 *Spike* coding sequence, since substitutions in the *Spike* protein are responsible for the vast majority of SARS-CoV-2 evolution and for driving SARS-CoV-2 immune escape[7,42,43]. In our model, we break down viral evolution into four essential steps: (i) determining the number of nucleotide substitutions happening in a time step; (ii) distributing substitutions between lineages within infected individuals; (iii) assigning the positions in the genome (location of nucleotide) where these take place; and (iv) choosing the nucleotide substitution (which nucleotide will substitute the previous one, e.g. A → T).

We modeled the total number of substitution events as a homogeneous Poisson process with expectation value: $\lambda(t) = NSR \cdot \Delta t \cdot L \cdot \sum_{i=1}^{|I(t)|} |l(t)|_i$, where $\Delta t$ denotes the time step to the next epidemiological event, $NSR$ is the

nucleotide substitution rate, $|l(t)|_i$ the number of lineages hosted by individual $i$, $L$ the genome length and $|I(t)|$ being the number of infected individuals. We assume that individuals within the population have the same likelihood of hosting a substitution per time step and consequently, we assign substitutions to infected individuals at random (uniform). While our model is able to utilize site-specific substitution rates (which can be derived from phylodynamics analysis) and any nucleotide substitution model, in the experiments presented here, we assumed uniform rates across the genome and utilized the Jukes-Cantor model[44] that assigns transitions and transversions probabilities as shown:

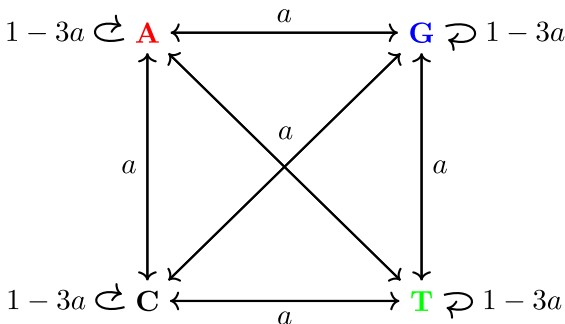

and can be described by a transition matrix. The transition matrix P(t) gives the probability that a specific substitution occurs at a site:

$$P(t) = \begin{pmatrix} P(A \to A) & P(A \to G) & P(A \to C) & P(A \to T) \\ P(G \to A) & P(G \to G) & P(G \to C) & P(G \to T) \\ P(C \to A) & P(C \to G) & P(C \to C) & P(C \to T) \\ P(T \to A) & P(T \to G) & P(T \to C) & P(T \to T) \end{pmatrix}$$

### Intra-host viral diversification

There is evidence that during acute SARS-CoV-2 infections, sometimes minor intra-host genetic lineages emerge with a frequency sufficient for transmission[45]. Depending on the duration of infection, the virus can establish distinct populations of intra-host lineages that may cross the threshold to become transmissible. This process is modeled by a third propensity:

$$a_3 = k_v \cdot \sum_{i=0}^{19} |x_i(t)| \qquad (3)$$

where $k_v$ is the intra-host lineage emergence rate, and the summation term refers to each infected individual. When this reaction happens, a new lineage is introduced in a randomly selected individual as a copy of an already existing lineage within that host; this models intra-host lineage emergence, a process that is independent of inter-host selective pressures. The lineages then proceed to evolve independently under the evolutionary model described above. To capture the observed heterogeneity of within-host diversification without imposing strong assumptions about its underlying biology, we assume that the rate of intra-host diversification is constant until a maximum number of dominant intra-host lineages is reached within an individual. For each host, this maximum is drawn from a uniform distribution, and we adopt an upper limit of five lineages so as not to constrain the model with conservative assumptions. Users can adjust this intra-host lineage limit (and distribution) as needed for specific modeling scenarios.

Once an individual has reached the maximum number of dominant intra-host lineages, subsequent lineage emergence events trigger the replacement of one existing lineage by a newly introduced one. This mechanism represents the competitive de-selection of intra-host lineages, driven by within-host competition, and operates independently of the lineages'

transmission fitness. A sensitivity analysis with regard to the intra-host lineage emergence rate $k_v$ is shown in Supplementary Fig. S3.

## Phenotypic model

We assign a relative transmission fitness score to each virus lineage present in the population at time $t$. In the model, an infecting virus variant is chosen according to its relative transmission fitness $p(l, t) = f(l, t)/\sum_i f(l, t)$. Notably, the transmission fitness of a variant will change over time and entails a part representing competition among lineages, and a non-competition term. The competition term represents the variants' ability to re-infect individuals, while the non-competition term is related to infection of virus-naive individuals, where we assume that virus lineages are equally capable of infecting individuals who had never been infected. By contrast, lineages need to be distinct from past lineages in order to re-infect individuals who had already been exposed to the virus.

$$f(l, t) = \underbrace{\pi_{\text{non-inf}}(t) \cdot (|l(t)|)^{-1}}_{\text{first infection}} + \underbrace{\pi_{\text{inf}}(t) \cdot d(l(t), c)}_{\text{re-infection}} \qquad (4)$$

where $\pi_{\text{inf}}(t) = \min\left(\frac{|D(t)|+|R(t)|}{|S(t)|}, 1\right)$ denotes the proportion of the population that has exited the infected compartment by time $t$ and $\pi_{\text{non-inf}} = \max\left(\frac{|S(t)|-(|D(t)|+|R(t)|)}{|S(t)|}, 0\right)$ denotes the proportion of infection-naive individuals. The variable $(|l(t)|)$ in the equation above denotes the number of viral lineages prevalent at time $t$. The function $d(l(t), c)$ denotes a measure of antigenic distance to some past immunity-inducing viral population. In the experiments, we test two scenarios for $d(l(t), c)$:

**Baseline model (linear).** In the baseline model, fitness is computed as the Hamming distance to the founder virus (wild-type). This assumes that as the virus evolves, new lineages emerge that are better at re-infecting individuals. While not biologically realistic for SARS-CoV-2, we utilize it as a baseline scenario to compare with when not considering infection history in the simulations. Therefore, we define

$$d(l(t), c_0) = \text{Hamming}(\text{lineage}, \text{consensus}(t_0))$$

i.e. the Hamming distance between the lineage within an individual and the founder sequence for the simulation.

**Immune waning model.** The second model accounts for acquired lineage-specific neutralizing immunity in the population and is related to mechanistic approaches of estimating relative lineage fitness in heavily immunized populations based on infection (and vaccination) history[7,43]. In this model, the relative fitness of viral lineages is assigned as the Hamming distance to the temporally weighted consensus sequence of all virus lineages that circulated in the population in the previous months. Past consensus sequences are weighed by a pharmacokinetic function describing the waning of antibodies in previously exposed individuals[7]. The weights for a consensus sequence that was prevalent $s \in [0, 180]$ days ago are calculated as

$$w(t,s) = \frac{e^{-k_e(t-s)} - e^{-k_a(t-s)}}{e^{-k_e(t_{\max}-s)} - e^{-k_a(t_{\max}-s)}} \qquad (5)$$

which denotes a normalized pharmacokinetic Bateman function where $t$ is the current time point and $k_e$ and $k_a$ are the antibody elimination and absorption rates (1/day), respectively. In the immune waning model, we define

$$d(l(t), \bar{c}) = \text{Hamming}(\text{lineage}, \text{consensus}_w eighted) \qquad (6)$$

In this model, we thus introduce a mechanism of interaction between the emerging lineages and the waning immunity in the population, which adds the time dimension to the phenotypic model and assumes that SARS-CoV-2 evolves to maximize its ability to infect an immunologically-experienced population[7].

## Numerical implementation

To run SIMPLICITY simulations, we adapted the Extrande algorithm[40] to our reaction network with time-varying propensities. Extrande enables exact stochastic simulation of such networks by overcoming the limitations of SSA, which assumes constant reaction propensities between events. Extrande introduces a virtual reaction (a reaction that does not change the state of the system) with a time-dependent propensity chosen such that the total propensity of the augmented system is piecewise constant (ensured by the UPPERBOUND, which is an upper bound on the sum of all the system's reaction rates). Reactions associated with the virtual reaction are discarded ('thinning' step). This permits correct numeric sampling of waiting times that are not exponentially distributed. We extended the Extrande algorithm, adding steps for intra-host state evolution, mutation, and fitness updates. Below is the pseudocode of the core SIMPLICITY simulation algorithm. Note that the LOOKAHEAD time horizon in this implementation is simply the final time of the simulation.

**Algorithm 1**. Extrande Core Loop in SIMPLICITY

```
 1: Initialize time t ← t₀, accumulator Δt_acc ← 0
 2: while t < t_final do
 3:     L ← LOOK_AHEAD(t, t_final)
 4:     B ← COMPUTE_UPPERBOUND(population)
 5:     Δt ~ Exp(1/B)
 6:     if Δt > L then                        ▷ Reject (leap) step
 7:         t ← t + L
 8:         UPDATE_TIME(population, t)
 9:         Δt_acc ← Δt_acc + L
10:         if Δt_acc ≥ δ_min then
11:             UPDATE_STEP(population, Δt_acc)
12:             MUTATION_STEP(population, Δt_acc)
13:             Δt_acc ← 0
14:         end if
15:     else                                  ▷ Accepted (reaction) step
16:         t ← t + Δt
17:         UPDATE_TIME(population, t)
18:         Δt_acc ← Δt_acc + Δt
19:         if Δt_acc ≥ δ_min then
20:             UPDATE_STEP(population, Δt_acc)
21:             MUTATION_STEP(population, Δt_acc)
22:             Δt_acc ← 0
23:         end if
24:         UPDATE_FITNESS_STEP(population)
25:         reaction_id ← REACTION_STEP(population, B)
26:     end if
27:     REPORTER.UPDATE(population, Δt, reaction_id, event_type)
28:     POPULATION.UPDATE_TRAJECTORY
29:     if day has advanced then
30:         POPULATION.UPDATE_LINEAGE_FREQUENCY_T(t)
31:     end if
32:     if CHECK_STOP_CONDITIONS(population, t) then
33:         break
34:     end if
35: end while
36: REPORTER.CLOSE
```

## Simulated trees

SIMPLICITY can provide the full infection and phylogenetic tree of a simulation. Infection trees are created by building a time-oriented binary tree (branch length is the difference from the time of getting infected to infecting a new individual) in which leaves are individuals and internal nodes represent infection events. Phylogenetic trees are binary trees (branch

length either genetic distance (Hamming) or time of emergence) in which each leaf is a lineage and internal nodes are substitution events. Lineages are defined by a unique set of substitutions, meaning that every sequence differing by at least 1 position in the genome from the others is defined as a separate lineage.

## Model parametrization

Each model was parameterized to ensure that the simulations could reproduce the population and viral evolutionary dynamics observed during the SARS-CoV-2 pandemic. Some of the parameters were derived from data collected during the pandemic, while others were taken from literature or manually fine-tuned for the purposes of this paper.

**Intra-host model.** Model parameters were taken from the already fitted published model[4].

**SIRD model.** The infection rate $\beta(t)$ is related to the virus reproduction number (R) as follows:

$$R = \beta(t) \cdot |S(t)| \cdot \tau_{\text{infectious}} \qquad (7)$$

The reproduction number of the virus, i.e., the average number of new cases generated during an individual's infectious period, is given by the product of the infection rate $\beta(t)$ and the number of susceptibles in the system scaled by the expected duration of infectiousness $\tau_{\text{infectious}}$. The relation between $R$ and $\beta$, leaves $R$ as free parameter, by solving for $\beta$ and setting $\tau_{\text{infectious}} = 11.41$ days[4], which can be specified for each simulation.

The diagnosis rate parameter $k_d$ can be estimated from a given diagnosis probability $P(\text{diag})$ by calculating the limit distribution

$$\boldsymbol{p}_{\infty}(\boldsymbol{x}) = \lim_{t \to \infty} \exp(\boldsymbol{B}t)\boldsymbol{p}_0(\boldsymbol{x}).$$

The last entry of $\boldsymbol{p}_{\infty}(\boldsymbol{x})$ denotes the distribution of diagnosed individuals and therefore the diagnosis probability $P(\text{diag})$. We set the default diagnosis probability to 0.1. In the simulations, diagnosed individuals are sequenced with a user-defined probability. The sequencing rate allows the user to decide which proportion of diagnosed cases will be stored as sequences (default = 0.05).

**Evolutionary model.** The nucleotide substitution rate (NSR), is estimated empirically, using a regression between phylogenetic distance (genetic distance from the phylogenetic tree root to each leaf, annotated with a genetic sequence) and sampling time of the sequence[26]. Our model follows the molecular clock hypothesis, such that genetic differences accumulate at a constant rate over time. Appropriate model parameterization would also allow reproducing inhomogeneous evolutionary rates across the tree. After each simulation, we use the simulated sequencing data to estimate the observed substitution rate (OSR).

SIMPLICITY allows the user to use any desired transition matrix $P(t)$. For the work presented here, we use a constant rate (0.33) for any transition or transversion. As the phenotype model we employ only uses the Hamming distance between sequences to assign a fitness value to a lineage, the genome composition has no impact on the transmission fitness, meaning that we can use these simplified assumptions for the scope of this paper.

**Phenotypic model.** The baseline phenotype model has no parameters. The parameters of the immune-waning model are set to $k_e = \frac{\ln(2)}{t_{\text{half}}}$, $t_{\text{half}} = 30$ and $t_{\text{max}} = 21$, according to previous work[7]. The antibody generation rate constant $k_a$ is obtained numerically by solving

$$t_{\text{max}} = \frac{\ln(k_a/k_e)}{k_a - k_e},$$

for $k_a$ using a numerical method.

**Table 1 | Parameters used in OSR tuning (Fig. 2) and phenotype model comparison experiments (Figs. 3, 4)**

| Parameter | OSR tuning | Phenotype model comparison |
|---|---|---|
| Population size | 1000 | 1000 |
| Infected individuals at start | 10 | 10 |
| Final time (days) | 1095 | 1095 |
| $\tau_3$ | 7.5 | 7.5 |
| $R$ | 1.1 | 1.1 |
| Diagnosis rate | 0.1 | 0.1 |
| IH virus emergence rate ($k_v$) | 0 | 0 |
| Nucleotide substitution rate (e) | $[10^{-6}, 3 \times 10^{-4}]$ | 0.00008759 |
| Phenotype model | immune_waning | baseline, immune_waning |
| Sequencing rate | 0.05 | 0.05 |
| Seeded simulations number | 100 | 300 |

## Experiments

The experiments presented in this paper showcase the use of SIMPLICITY to investigate of SARS-CoV-2 intra-host and population-level dynamics and their interplay. Table 1 resumes the parameter values used for each experiment. We define an experiment as a set of $n$ simulations with a set of fixed and varying parameters. The first experiment (Fig. 2) goal is to tune the model to a range of observed substitution rates (OSR) that correspond to real-world SARS-CoV-2 data. We modeled the SARS-CoV-2 *Spike* gene, as it is the main evolutionary driver. We ran 100 simulations per parameter set, fixing all parameters except the nucleotide substitution rates (NSR) (we used 15 logarithmically spaced values $\in [10^{-6}, 3 \times 10^{-4}]$) and used the resulting data to fit different functions (lin, log, exp, tan, spline) to the NSR/OSR relationship, using the Levenberg-Marquardt algorithm. We selected the best-fitting curve by comparing the resulting Akaike Information Criterion.

In the second experiment (Figs. 3 and 4), we ran 300 simulations for each phenotype model (baseline, immune waning) and compared the simulation results (system trajectories, lineage frequency dynamics, number of selective sweeps happening, $R$ effective, infection and phylogenetic trees, and entropy). The plots shown in Fig. 3 used the simulation raw output data, $R$ effective, and the true phylogenetic tree reconstructed from the evolution histories of the lineages. $R$ effectiveness was computed by calculating the ratio of births to death events over a sliding time window of 21 days. For the average population $R$ effective, we define birth events as an individual becoming infectious and a death event as an individual becoming non-infectious (entry in post-infectious phase or diagnosis and isolation). For lineage-specific $R$ effective, we define births as a lineage becoming infection capable (either when an infected individual becomes infectious or when a new lineage emerges in an infectious individual). Lineage death events are defined as lineage removal from the infectious individuals pool, either due to individuals becoming non-infectious or to evolution into a new lineage. The phylogenetic tree is reconstructed as described in the respective section above and plotted on a circular axis, colored by lineage. All lineages in a single simulation are mapped to a rainbow gradient color map that assigns a unique color to each lineage, ordered by the time of emergence. To count the number of selective sweeps that happen in a simulation, we use the lineage frequency data and define a sweep event as a lineage crossing the 50% frequency threshold and staying above it for at least 21 days. In Fig. 4, we show a violin plot comparing the distributions of selecting sweeps counts in simulations running the baseline vs the immune waning phenotype model. We compared the two groups using a Mann-Whitney two-sided test. Finally, we compare the lineage frequency entropy trajectories of each group to investigate how the phenotype model affects lineage diversification. In Fig. 4, we show the entropy data for a single simulation on the raw lineage

data. In Fig. S3, we also show the same analysis run on frequency data of clustered lineages (defining a lineage as a cluster of single-substitution lineages that share at least 5 mutations).

## Statistics and reproducibility

We used a Mann-Whitney two-sided statistical test in Fig. 4 with N=206. To reproduce the experiments presented here, the user can simply install SIMPLICITY 1.1.4[46] from the provided repository and then run scripts/experiments/00_generate_data_OSR_fit.py for the OSR fitting experiment (Fig. 2) and scripts/experiments/01_generate_data_SIMPLICITY_paper.py for the main experiment (Figs. 3, 4). To reproduce the figures, run the scripts/paper_figures scripts. For Fig. S3 (sensitivity with regard to virus emergence rate $k_v$), run scripts/experiments/generate_data_IH_lineages.py and scripts/plots/01_plot_IH_lineages.py.

## Reporting summary

Further information on research design is available in the Nature Portfolio Reporting Summary linked to this article.

## Data availability

The datasets generated and analyzed during the presented study are available in a publicly available Zenodo repository[47] (https://doi.org/10.5281/zenodo.17368796), together with the figures' source data.

## Code availability

The SIMPLICITY model user interface was implemented using Python and is available on GitHub under the GNU GPL v3 licence: https://github.com/PietroGx/SIMPLICITY. The version used for the work presented here is SIMPLICITY v.1.1.4[46]. The description of variables in the SIMPLICITY core algorithm is outlined in Supplementary Table S3.

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

## Acknowledgements

The authors thank Ariane Weber and Sanni Översti for insightful discussions on SARS-CoV-2 evolution and phylogenetics and Silvan Wehrli for coding advice. We thank Wiep van der Toorn for feedback regarding the SARS-CoV-2 intra-host model and to Nadezhda Malysheva for discussion on the numeric implementation. MvK and NG acknowledge funding by the Deutsche Forschungsgemeinschaft (DFG, German Research Foundation) under Germany's Excellence Strategy - The Berlin Mathematics Research Center MATH+ (EXC-2046/1, project ID: 390685689). This project was supported by Germany's Federal Ministry of Health (BMG) under grant no. 2523DAT400 (project "AI-assisted analysis and visualization of pandemic situations," AI-DAVis-PANDEMICS).

## Author contributions

Pietro Gerletti: software development, theoretical work and model development, experimental design, experiment realization, manuscript writing; Nils Gubela: theoretical work, oversight of mathematical formulation, manuscript writing; Jean-Baptiste Escudié: software development; Denise Kühnert: supervision, theoretical work, manuscript writing; Max von Kleist: supervision, theoretical work, model development, manuscript writing.

## Funding

## Competing interests

The authors declare no competing interests.
