## [Transparent Peer Review file · Communications Biology]

SIMPLICITY: an agent-based, multi-scale mathematical model to study SARS-CoV-2 intra- and between-host evolution.

Corresponding Author: Mr Pietro Gerletti

Version 0:

Reviewer comments:

Reviewer #1

(Remarks to the Author)

Review for "SIMPLICITY: an agent-based, multi-scale mathematical model to study SARS-CoV-2 intra- and between-host evolution"

Thank you to the editors and authors for the opportunity of reviewing this work.

In this paper, the authors present a software tool of their own design that combines evolutionary and epidemiological dynamics. The tool, named SIMPLICITY, is structured as a stochastic, agent-based simulation in which individual agent "hosts" can be infected by pathogens with specific sequences. Hosts can belong to one of a number of different epidemiological compartments (Susceptible, Recovered, Diagnosed, and a number of sequential Infected compartments). New genetic variants (strains) of pathogens arise within infected hosts at a certain rate following a statistical model of molecular evolution. Specific pathogens within infected hosts can infect other hosts that are susceptible to them based on their transmissibility, which is determined according to a user-specified model that maps pathogen genetic sequence to fitness (defined in terms of transmissibility). The authors construct SIMPLICITY to fit known epidemiological and evolutionary dynamics of SARS-CoV-2, and use their simulation platform to qualitatively recapitulate the selective sweep system behavior observed in SARS-CoV-2 evolution throughout the pandemic (particularly after its early years) by including immune waning in the simulation.

I must admit that I am biased given my own research on evo-epidemiological simulations, but I find the nature of the research presented here exciting and stimulating. SIMPLICITY joins a small but growing number of computational tools that aim to capture the interplay between pathogen evolutionary and epidemiological dynamics. Through the continued effort of teams such as the authors of this work, I am excited to see a small research community coalesce around methods development for evo-epidemiological simulation. The method described can be a valuable contribution to this line of work.

In what follows, I will make three main points followed by a number of minor comments. The major points I want to highlight about the manuscript are 1) its innovations with respect to previous techniques, 2) the nature of intrahost evolution in SIMPLICITY, and 3) the relevance of the results shown as applications of the method.

1. SIMPLICITY and other evo-epidemiological simulations

The fundamental structure of SIMPLICITY as a stochastic, agent-based evo-epidemiological simulation is similar to previous work such as the e3SIM (Xu et al., <https://doi.org/10.1101/2024.06.29.601123>) and Opqua (Cárdenas et al., <https://doi.org/10.1126/sciadv.abo0173> ; discussed in this manuscript and of which I am an author, for full disclosure) frameworks. Opqua in particular handles a similar simulation structure with molecular evolution parameters, user-defined phenotypic models mapping genetic sequence to fitness (defined as transmissibility among other variables), intrahost strain emergence, and interhost epidemiological spread. Although SIMPLICITY is more constrained in its intended usage and the range of biological phenomena it considers in its structure, it exhibits two main methodological innovations: a more complex consideration of disease progression across a number of infected stages or compartments and the usage of the Extranode simulation algorithm developed by some of the authors previously (Voliotis et al. <https://doi.org/10.1371/journal.pcbi.1004923>) as the basis of the simulation loop, as opposed to a Gillespie simulation algorithm. I find the usage of Extranode particularly interesting and a valuable contribution to the work being done in these kinds of simulations. I will certainly consider it in my own work! Beyond these differences in method, the manuscript

presented here applies SIMPLICITY to replicate real-world pathogen dynamics, something the work done in Cárdenas et al. does not attempt to do (although the work by Xu et al. with e3SIM does).

2. Intra-host fitness, evolution, and multiscale modeling in SIMPLICITY

The manuscript asserts that SIMPLICITY constitutes a multiscale model in which the interplay of intra-host and inter-host dynamics replicates mechanisms of pathogen evolution in order to make accurate predictions. This is an excellent objective to attempt. However, the way in which SIMPLICITY considers intra-host evolution is limited to only random emergence of mutant strains, regardless of their potential fitness. This is related to the fact that fitness, as defined in the model, is limited to inter-host transmissibility. This may be a reasonable first-order assumption for some disease systems such as many acute respiratory infections such as influenza or most of SARS-CoV-2 evolution. However, in some disease systems, evolution is driven by strong intra-host selection to avoid immune responses or treatment, as is the case with HIV, some bacterial and parasitic infections, and chronic cases of SARS-CoV-2 (which can give rise to so-called “saltation” variants). Although incorporating this kind of intra-host fitness and selection is probably outside of the scope of the manuscript in its current form, I believe that acknowledging this limitation is important, particularly when characterizing SIMPLICITY as a multiscale model: although events at multiple scales are being considered, selection is only being considered at the host population scale, not at the intra-host scale. Discussion of this limitation in the manuscript’s chosen application context of SARS-CoV-2 infection is also important, given the importance of chronic SARS-CoV-2 cases for the evolution of the virus (which the authors themselves point out).

3. Relevance of application results

The objective of this paper is to present a method first and foremost, rather than to make a claim about a biological phenomenon. In this sense, it is not an issue that the results presented here about SARS-CoV-2 evolution being driven by selective sweeps dependent on waning population immunity are by and large confirmatory. However, I wonder if the presentation of two alternative models, the linear fitness model and the immune waning one as an alternative, is necessary. The linear model may be viewed as a slight strawman argument, given that it does not correspond with biological mechanism: in an adaptive landscape driven by immune selection, increased distance from WT is simply an emergent behavior of selected strains as they evade acquired immunity, but there is no biological mechanism by which this would be the case other than the succession of acquired immune histories that constitutes the mechanistic basis of the second hypothetical model presented, the immune history/waning model. To put it differently, the first hypothesis is presented as an alternative, but may actually just be a simplified version of an emergent behavior of the second. Given the nature of SIMPLICITY as a mechanistic model, encoding mechanisms and observing emergent behaviors should be the desired outcome. In this light, I wonder if there is a better way to frame the discussion of the hypotheses, or different hypotheses with which to show SIMPLICITY’s power.

Besides the major points above, below I list some minor points to draw the authors’ attention to throughout the text.

36–50: This intra-host evolution is key in other pathogen systems as well, such as HIV and some bacterial infections.

84: Perhaps “specifically within the context of SARS-CoV-2” should be added to the end of this sentence.

174: Perhaps a new lineage can be introduced as a copy with a mutation already on it (at a net rate that accounts for the mutation rate). Otherwise, the rate of lineage diversification has no empirical analogue: in a real infection, a diversification or lineage emergence event is characterized from the point in which a lineage’s last common mutation is acquired. This has the benefit of additional computational efficiency, since one fewer event is considered.

180–182: What is the rationale behind having a maximum number of intra-host lineages, and behind having it be a random number that varies according to host? Presumably this number might be proportional to some maximum viral load or carrying capacity that may vary by individual host according to its immune system. However, would this then have an effect on transmission rate, going contrary to the next statement made on lines 182–183?

229: Citing the Voliotis et al. (2016) paper describing Extranade directly in this section would be practical for the reader. Regarding Algorithm 1, it would be helpful to have a quick definition or summary of what the LOOK_AHEAD and COMPUTE_UPPERBOUND functions are doing, although I understand that they are discussed in Voliotis et al. (2016) under different terminology.

484–487: As the lead developer of Opqua, thanks for discussing our work! Here are some potentially useful additional points: although I would not say that the framework focuses on evolution across fitness valleys (that was just the application case), another significant advantage of SIMPLICITY over Opqua as published currently is the size of simulations. Opqua simulations with more than 105 individual hosts rapidly become computationally intractable without advanced hardware, something that SIMPLICITY is better equipped to achieve (at least by one order of magnitude, according to your discussion).

487: An additional framework the users should find relevant to discuss and compare is e3SIM (<https://doi.org/10.1101/2024.06.29.601123>).

498–500: These would be good options of example applications that can highlight the novelty or applicability of the method

532–534: I would be uneasy assuming all individuals imprint on the same virus. There is demographic turnover to account for at the timescale of SARS-CoV-2 evolution, so using a simple, single distance from WT metric to derive antigenic escape-determined fitness throughout the population would not be adequate.

537–541: Although I agree accounting for infection history could be a significant improvement for deep learning models of antigenic escape, an argument could also be made that these models already account for this implicitly based on the training dataset. The fact that a mechanistic model like SIMPLICITY corroborates the importance of infection history does not necessarily imply that these non-mechanistic, purely statistical machine learning models must explicitly account for infection history.

565–569: I agree that simulating chronic infections would be an interesting application! To do this, it would be important to incorporate methods to allow for intrahost selection and evolution, as explained in one of the main points.

Figure 2: Perhaps showing a measure of goodness of fit would be important here.

Figure 4: Comparing these entropy results from the simulated data to what is calculated from real-world observational data from the pandemic would be valuable.

Thank you for sharing your interesting work! I hope we have the opportunity to share notes some day.

Best wishes,

Pablo

Pablo Cárdenas Ramírez
Postdoctoral Research Fellow,
Ragon Institute of MIT, Harvard, & MGH
Incoming Assistant Professor
R.F. Smith School of Chemical & Biomolecular Engineering, Cornell University

Reviewer #2

(Remarks to the Author)

In this study the authors investigated the viral evolution and population spread of SRAS-CoV-2 using a multi-scale mathematical model that combines within-host disease progression and viral evolution, with a population-level model of virus transmission and immune evasion. Although it is an interesting work, there are many parts not clearly explained.

Why was the model named as “SIMPLICITY”?

SIDR model or SIRD model?

A brief introduction of “rejection-based exact stochastic simulation method” should be given.

Intra-host model: how are the five different phases and 20 compartments related? In matrix A, what are these parameters: r_1, r_2, r_3, r_4 ? how are x_0, \dots, x_{19} classified? Why “an individual can transmit if infection has progressed to x_5, \dots, x_{18} ”?

Equation (2): does it suggest that individuals in each of the infection phase from x_6 to x_{19} can be diagnosed with the same rate? However, Fig 1 shows that compartments x_4 and x_5 can also be diagnosed, and X_{19} (yellow) is already diagnosed.

As matrix A, Matrix B is also unclear.

“After an individual recovers, a new susceptible is introduced in the population pool, keeping the sum of infected + susceptible constant during a simulation (reinfection).”

Does this imply that after recovery, the individual quickly becomes entirely susceptible. This is surely not true for SARS-CoV-2 infection. Duration of immunity needs to be included.

Line: 152: what are “infectious agents”? please specify “positions in the genome”?

Line 153: define “nucleotide substitution”

Lines 156-7: “e is the nucleotide substitution rate” Better to avoid using e as any parameter because e is usually assigned for the natural constant.

Line 165: Briefly introduce Jukes-Cantor model.

Equation (3): “each infected individual” Is x_0 infected? Fig 1 shows that x_0 is in white, --susceptible?

Line 175: what does “the reaction fires” imply?

Lines 178-181: “We assume that the rate of intra-host diversification is constant and not depending on the number of lineages already present inside the host. Each individual has a maximum number of lineages that they can host” “not depending on” and “has a maximum number” cannot be true at the same time.

Line 182: why 5?

Line 187: how to model “within-host competition”?

Lines 190-192: The authors seem to argue this in two mutual ways. which way is used or workable?

Line 195: “virus lineages are equally capable at infecting...”, why? does this assume that all the lineages are of the same transmission fitness? This is not in agreement with “This assumes that as the virus evolves, new lineages emerge that are

better at infecting new individuals." (Linear fitness model)

Line 199: the expression given is not "the proportion of the population that had been infected by time t" The correct one should be $(D(t)+R(t)+I(t))/N(t)$ with $N(t)$ representing the total number of individuals in the population.

Equation for $d(I(t),c_0)$ needs to be explained.

Line 215: why six months?

Equation (5): what are the values of parameters k_e and k_a ?

Lines 224-5: some details of "mechanism of interaction..." should be specified.

Line 233: what is SSA?

Line 237: what is "virtual reaction"?

In the table for "Algorithm 1...", add a table to list the definitions of all variables used.

Line 254: "Lineages are defined by a unique set of substitutions." What is a unique set?

Line 290: what is transition matrix $P(t)$?

Lines 310-11: "fixing all parameters except the nucleotide substitution rates" The values of all other parameters should be given. If there are any uncertainties in other parameters, how to confirm the robustness of the conclusion?

Lines 322-4: "R effective was computed by calculating the ratio of births to deaths events over a sliding time windows of 21 days." Does this calculation agree with equation (7)?

Reviewer #3

(Remarks to the Author)

Please see attached comments

Version 1:

Reviewer comments:

Reviewer #1

(Remarks to the Author)

I'm grateful to the authors for their attention to the comments and suggestions! By and large, I believe the authors acknowledge the scope and limitations of the work more accurately. Three small suggestions come to mind:

1. In line 592, the number "105" comes from a formatting error from a previous comment of mine, which originally conveyed "10⁵" using superscript notation, but was rendered on the PDF incorrectly.

2. In line 597: This might be true about eSIM3's inability to model intrahost evolution, but I would argue it is also true about SIMPLICITY, as I argued in point #2 of the review. In this sense, bringing it up in this context might not be the differentiating factor that you are looking for.

3. Regarding the last point on the review, which I will quote in full below—discussing this in the manuscript could be useful to illustrate the limitations and future directions!

"Figure 4: Comparing these entropy results from the simulated data to what is calculated from real-world observational data from the pandemic would be valuable.

This is definitely something we have considered but decided against as the evolutionary model is simplified compared to real data and we are not considering other aspects of SARS-CoV-2 evolution that definitely affected what happened in the real world during the pandemic. Once long-shedders are added in the simulations and a more realistic evolutionary model is used the comparison with real world entropy data would make more sense and provide a valuable insight. Additionally, we could not find any entropy related studies that would be a sensible comparison for our model."

Thank you to the authors for their work and I look forward to more scientific interactions with them!

Pablo Cárdenas R.

Reviewer #3

(Remarks to the Author)

I have reviewed the changes made by the authors in response to my comments and the comments of the other two reviewers, and am satisfied with their responses. As always, there is more work to be done and more scenarios to consider and model but I think this manuscript is in a good state and there is no point in drawing out the review and revision process on a satisfactory manuscript! I will be excited to see it published.

Response letter

Title: SIMPLICITY: an agent-based, multi-scale mathematical model to study SARS-CoV-2 intra- and between-host evolution

We would like to thank the referees for their valuable and insightful feedback on our manuscript. We believe that their suggestions have greatly contributed to improving its quality. Each reviewer's comments and concerns have been addressed, with corresponding revisions made to the manuscript, when applicable. Below, we provide the reviewers' comments alongside our responses.

Referee #1:Mathematical Modeling; epidemiology and evolution

Review for "SIMPLICITY: an agent-based, multi-scale mathematical model to study SARS-CoV-2 intra- and between-host evolution"

Thank you to the editors and authors for the opportunity of reviewing this work.

In this paper, the authors present a software tool of their own design that combines evolutionary and epidemiological dynamics. The tool, named SIMPLICITY, is structured as a stochastic, agent-based simulation in which individual agent "hosts" can be infected by pathogens with specific sequences. Hosts can belong to one of a number of different epidemiological compartments (Susceptible, Recovered, Diagnosed, and a number of sequential Infected compartments). New genetic variants (strains) of pathogens arise within infected hosts at a certain rate following a statistical model of molecular evolution. Specific pathogens within infected hosts can infect other hosts that are susceptible to them based on their transmissibility, which is determined according to a user-specified model that maps pathogen genetic sequence to fitness (defined in terms of transmissibility). The authors construct SIMPLICITY to fit known epidemiological and evolutionary dynamics of SARS-CoV-2, and use their simulation platform to qualitatively recapitulate the selective sweep system behavior observed in SARS-CoV-2 evolution throughout the pandemic (particularly after its early years) by including immune waning in the simulation.

I must admit that I am biased given my own research on evo-epidemiological simulations, but I find the nature of the research presented here exciting and stimulating. SIMPLICITY joins a small but growing number of computational tools that aim to capture the interplay between pathogen evolutionary and epidemiological dynamics. Through the continued effort of teams such as the authors of this work, I am excited to see a small research community coalesce around methods development for evo-epidemiological simulation. The method described can be a valuable contribution to this line of work.

In what follows, I will make three main points followed by a number of minor comments. The major points I want to highlight about the manuscript are

- 1) its innovations with respect to previous techniques,
- 2) the nature of intrahost evolution in SIMPLICITY, and
- 3) the relevance of the results shown as applications of the method.

1. SIMPLICITY and other evo-epidemiological simulations

The fundamental structure of SIMPLICITY as a stochastic, agent-based evo-epidemiological simulation is similar to previous work such as the e3SIM (Xu et al., <https://doi.org/10.1101/2024.06.29.601123>) and Opqua (Cárdenas et al., <https://doi.org/10.1126/sciadv.abo0173> ; discussed in this manuscript and of which I am an author, for full disclosure) frameworks. Opqua in particular handles a similar simulation structure with molecular evolution parameters, user-defined phenotypic models mapping genetic sequence to fitness (defined as transmissibility among other variables), intrahost strain emergence, and interhost epidemiological spread. Although SIMPLICITY is more constrained in its intended usage and the range of biological phenomena it considers in its structure, it exhibits two main methodological innovations: a more complex consideration of disease progression across a number of infected stages or compartments and the usage of the Extrade simulation algorithm developed by some of the authors previously (Voliotis et al. <https://doi.org/10.1371/journal.pcbi.1004923>) as the basis of the simulation loop, as opposed to a Gillespie simulation algorithm. I find the usage of Extrade particularly interesting and a valuable contribution to the work being done in these kinds of simulations. I will certainly consider it in my own work! Beyond these differences in method, the manuscript presented here applies SIMPLICITY to replicate real-world pathogen dynamics, something the work done in Cárdenas et al. does not attempt to do (although the work by Xu et al. with e3SIM does).

We thank the reviewer for the positive feedback. In the revision, we included the comments made by the reviewers that complement works described above, such as the consideration of within-host viral dynamics, the use of Extrade for simulation and the application to real-world pathogen dynamics.

2. Intrahost fitness, evolution, and multiscale modeling in SIMPLICITY

The manuscript asserts that SIMPLICITY constitutes a multiscale model in which the interplay of intrahost and interhost dynamics replicates mechanisms of pathogen evolution in order to make accurate predictions. This is an excellent objective to attempt. However, the way in which SIMPLICITY considers intrahost evolution is limited to only random emergence of mutant strains, regardless of their potential fitness. This is related to the fact that fitness, as defined in the model, is limited to interhost transmissibility. This may be a reasonable first-order assumption for some disease systems such as many acute respiratory infections such as influenza or most of SARS-CoV-2 evolution. However, in some disease systems, evolution is driven by strong intrahost selection to avoid immune responses or treatment, as is the case with HIV, some bacterial and parasitic infections, and chronic cases of SARS-CoV-2 (which can give rise to so-called “saltation” variants). Although incorporating this kind of intrahost fitness and selection is probably outside of the scope of the manuscript in its current form, I believe that

acknowledging this limitation is important, particularly when characterizing SIMPLICITY as a multiscale model: although events at multiple scales are being considered, selection is only being considered at the host population scale, not at the intrahost scale. Discussion of this limitation in the manuscript's chosen application context of SARS-CoV-2 infection is also important, given the importance of chronic SARS-CoV-2 cases for the evolution of the virus (which the authors themselves point out).

The reviewer is correct in pointing out that in its present form, SIMPLICITY simplifies intra-host dynamics and focuses solely on “transmission fitness” in an immunologically experienced population (i.e. mutations that may overcome immunity in the population). We agree that modeling detailed intra-host evolution is outside the scope of the current work. Our modeling choice is based on the assumption that for acute respiratory viruses like SARS-CoV-2, the selective pressures at the intra-host and inter-host levels can be considered independent processes (<https://doi.org/10.1093/molbev/msad204>). In particular, Ruan et al. provide evidence that within-host and between-host selective pressures act independently and possibly antagonistically in SARS-CoV-2. Notably, when with- and between host evolution is independent, ‘evolutionary directions’ are also independent and the *duration* of within-host evolution has consequences with regards to the magnitude of movement on the (transmission-)fitness landscape. Following the reviewer's suggestion, we have expanded the discussion in the manuscript (line 624-632). We now acknowledge this limitation, clarifying that selection is considered at the host population scale rather than the intra-host scale. As the reviewer correctly points out, this limits the scope of interpretation, too: We would not be able to detect evolutionary marks in the genome, such as APOBEC3-mediated marks of genomic editing found in the Mpox virus (MPXV).

3. Relevance of application results

The objective of this paper is to present a method first and foremost, rather than to make a claim about a biological phenomenon. In this sense, it is not an issue that the results presented here about SARS-CoV-2 evolution being driven by selective sweeps dependent on waning population immunity are by and large confirmatory. However, I wonder if the presentation of two alternative models, the linear fitness model and the immune waning one as an alternative, is necessary. The linear model may be viewed as a slight strawman argument, given that it does not correspond with biological mechanism: in an adaptive landscape driven by immune selection, increased distance from WT is simply an emergent behavior of selected strains as they evade acquired immunity, but there is no biological mechanism by which this would be the case other than the succession of acquired immune histories that constitutes the mechanistic basis of the second hypothetical model presented, the immune history/waning model. To put it differently, the first hypothesis is presented as an alternative, but may actually just be a simplified version of an emergent behavior of the second. Given the nature of SIMPLICITY as a mechanistic model, encoding mechanisms and observing emergent behaviors should be the desired outcome. In this light, I wonder if there is a better way to frame the discussion of the hypotheses, or different hypotheses with which to show SIMPLICITY's power.

We agree that presenting the linear fitness model as an alternative biological hypothesis was not optimal, as it, at best, be a coincidental consequence of a mechanistic process, for example when immunity does not wane-off and incidence is roughly constant. In the revised manuscript, we have renamed it baseline model and clarified that it serves as a simplified reference scenario used to illustrate the system's behavior in the absence of immune waning. This highlights that the immune-waning model encodes the mechanistic processes driving immunity-mediated selection, while the baseline model provides a point of comparison for understanding the emergent differences in evolutionary dynamics.

Besides the major points above, below I list some minor points to draw the authors' attention to throughout the text.

36–50: This intrahost evolution is key in other pathogen systems as well, such as HIV and some bacterial infections.

Added in text (line 49-52)

84: Perhaps “specifically within the context of SARS-CoV-2” should be added to the end of this sentence.

Done, line 87

174: Perhaps a new lineage can be introduced as a copy with a mutation already on it (at a net rate that accounts for the mutation rate). Otherwise, the rate of lineage diversification has no empirical analogue: in a real infection, a diversification or lineage emergence event is characterized from the point in which a lineage's last common mutation is acquired. This has the benefit of additional computational efficiency, since one fewer event is considered.

We value the reviewer's point. However, we are hesitant to implement this change because it would introduce additional complexity when dealing with more detailed evolutionary models. In the current implementation, the process of intra-host lineage diversification is separated into two steps: **(1)** the emergence of an intra-host lineage and **(2)** the independent evolutionary model that acts on the newly emerged virus. At present, we treat intra-host viral emergence as an explicit Extrande reaction with a user-defined rate, while the evolutionary model operates as a separate process that is executed in bulk for computational efficiency. In future versions of SIMPLICITY, we plan to develop a more realistic intra-host evolutionary model, in which the intra-host viral emergence rate will have a clear empirical analogue. This will allow us to capture the complexity of intra-host diversification without compromising the clarity and flexibility of the current modeling framework. We updated the intra-host viral diversification paragraph.

180–182: What is the rationale behind having a maximum number of intrahost lineages, and behind having it be a random number that varies according to host? Presumably this number

might be proportional to some maximum viral load or carrying capacity that may vary by individual host according to its immune system. However, would this then have an effect on transmission rate, going contrary to the next statement made on lines 182–183?

SARS-CoV-2 infections with establishment co-occurring intra-host lineages have been observed, in some cases due to different ecological niche colonization (e.g. different organs / tissues), or due to a compromised immune system (long-shedders), which is the subject of ongoing work. The reasoning for a maximum number of co-occurring lineages is motivated, as the reviewer correctly points out, by some maximum host-specific virus load, and a threshold frequency that has to be surpassed by lineages to be selected for transmission. In the work presented here we used a uniform distribution to assign the maximum lineage hosting capacity of individuals so as not to impose any hard assumption on the underlying biology of the process. In the results presented in this paper, we set the virus duplication rate to 0, which implicitly assumes that, at the time point of onward transmission, the within-host virus population has not diverged into several evolutionary branches. This is in line with the vast majority of transmission events, which occur within days of infection and ultimately result in little evolution between direct transmission pairs (PMID: 35853960, PMID: 35062291). We nevertheless include this mechanism in SIMPLICITY as it is relevant in other possible scenarios. We modified and corrected the paragraph in the paper to clarify this.

229: Citing the Voliotis et al. (2016) paper describing Extranode directly in this section would be practical for the reader. Regarding Algorithm 1, it would be helpful to have a quick definition or summary of what the LOOK_AHEAD and COMPUTE_UPPERBOUND functions are doing, although I understand that they are discussed in Voliotis et al. (2016) under different terminology.

We added the citation and a short clarifier for both UPPERBOUND and LOOKAHEAD (line 280-295) and a table explaining all variables in the algorithm box in the supplementary.

484–487: As the lead developer of Opqua, thanks for discussing our work! Here are some potentially useful additional points: although I would not say that the framework focuses on evolution across fitness valleys (that was just the application case), another significant advantage of SIMPLICITY over Opqua as published currently is the size of simulations. Opqua simulations with more than 105 individual hosts rapidly become computationally intractable without advanced hardware, something that SIMPLICITY is better equipped to achieve (at least by one order of magnitude, according to your discussion).

Thank you. Modified accordingly in the discussion (line 552 - 556)

487: An additional framework the users should find relevant to discuss and compare is e3SIM (<https://doi.org/10.1101/2024.06.29.601123>).

Added (line 556-559)

498–500: These would be good options of example applications that can highlight the novelty or applicability of the method

While we agree that this would be a good example application, we believe them to be outside the scope of the current publication. Setting up this modelling scenario is doable but would add more content to an already quite long and heavy paper. We therefore decided to limit the content of the current publication to ensure better readability and clearer presentation.

532–534: I would be uneasy assuming all individuals imprint on the same virus. There is demographic turnover to account for at the timescale of SARS-CoV-2 evolution, so using a simple, single distance from WT metric to derive antigenic escape-determined fitness throughout the population would not be adequate.

We totally agree with the reviewer and had previously used this paragraph to discuss how to interpret AI models that predict “viral escape” without considering immune history (see also next comment). Following a slight reshaping (less focus on AI models), we removed this section of the manuscript with less focus being put on the discussion of AI models and more focus on discussing the main hypotheses of our work, as well as related works..

537–541: Although I agree accounting for infection history could be a significant improvement for deep learning models of antigenic escape, an argument could also be made that these models already account for this implicitly based on the training dataset. The fact that a mechanistic model like SIMPLICITY corroborates the importance of infection history does not necessarily imply that these non-mechanistic, purely statistical machine learning models must explicitly account for infection history.

565–569: We agree with the point made by the reviewer regarding the fact that we cannot, from SIMPLICITY alone, make statements regarding the importance of infection history. Our statement was also referring to related works (PMID: 37875109, PMID: 39880955) that clearly show that the history of antigen exposure (infection & vaccination) and immune waning determine selective pressures and thus steer evolutionary trajectories at the population level. We disagree however with the statement that machine learning models implicitly account for infection history for two reasons: Some machine learning models only use pre-pandemic data (i.e. PMID: 37821700, which uses EVE (PMID: 34707284) underneath), such as homologous sequences (comparison across the coronavirus family). Other machine learning models may use molecular surveillance data (sequence data), but these do not reflect infection history: Evolutionary trajectories differ quite substantially between countries and continents (i.e. investigated herein: PMID: 39880955), such that there is no ‘global infection history’ (just think of ‘Alpha’ spread in Europe, US, ‘Beta’ in Africa, ‘Gamma’ in South America). At the same time, availability of sequencing data strongly differs between the global north and south and hence training on globally available sequencing data does certainly not reflect infection history in any particular setting and may moreover introduce some strong biases towards settings in which more data is available. In the paper, we removed the discussion of AI models as mentioned before as it is outside the scope of the manuscript.

I agree that simulating chronic infections would be an interesting application! To do this, it would be important to incorporate methods to allow for intrahost selection and evolution, as explained in one of the main points.

Thank you for the feedback. We will consider this for future work on chronic infections!

Figure 2: Perhaps showing a measure of goodness of fit would be important here.

Added to Supplementary.

Figure 4: Comparing these entropy results from the simulated data to what is calculated from real-world observational data from the pandemic would be valuable.

This is definitely something we have considered but decided against as the evolutionary model is simplified compared to real data and we are not considering other aspects of SARS-CoV-2 evolution that definitely affected what happened in the real world during the pandemic. Once long-shedders are added in the simulations and a more realistic evolutionary model is used the comparison with real world entropy data would make more sense and provide a valuable insight. Additionally, we could not find any entropy related studies that would be a sensible comparison for our model.

Referee #2:infectious disease dynamics; population genetics

In this study the authors investigated the viral evolution and population spread of SARS-CoV-2 using a multi-scale mathematical model that combines within-host disease progression and viral evolution, with a population-level model of virus transmission and immune evasion. Although it is an interesting work, there are many parts not clearly explained.

Why was the model named as “SIMPLICITY”?

Line 71: Stochastic simulation of sars-cov-2 spreading and evolution accounting for within-host dynamics (SIMPLICITY). It is an acronym for the above and, at the same time, we found it to be a bit humorous. (and hope this is not perceived differently)

SIDR model or SIRD model?

SIRD, corrected, thanks for pointing out!

A brief introduction of “rejection-based exact stochastic simulation method” should be given.

That is an important point, yes. Added a concise description at the beginning of the Methods section (line 97-107)

Intra-host model: how are the five different phases and 20 compartments related? In matrix A, what are these parameters: r_1, r_2, r_3, r_4 ? how are x_0, \dots, x_{19} classified? Why “an individual can transmit if infection has progressed to x_5, \dots, x_{18} ”?

We added explanations regarding the parameters and design of the intra host model and the biological meaning of the compartments (line 110 - 158). We apologize if this was confusing! In brief, we used a previously developed descriptive, stochastic model of within host dynamics, that reflects population variability with regards to the timeline of infectiousness, test positivity, onset and duration of symptoms, etc. (PMID: 33899034). The model was parameterized with clinical and ex vivo data and (later in the pandemic) validated with data from the human challenge study (PMID: 35361992), when it became available.

A sketch explaining the model from the previous publication is depicted below along with data used to parameterize it.

Equation (2): does it suggest that individuals in each of the infection phase from x_6 to x_{19} can be diagnosed with the same rate? However, Fig 1 shows that compartments x_4 and x_5 can also be diagnosed, and x_{19} (yellow) is already diagnosed.

We apologize if the description of the model caused confusion. In principle, the reviewer is correct. However, as an infected individual moves through the intra-host model, the probability to actually be in one of the compartments where infection is detectable rapidly decreases (as shown above, lower left panel). This reflects actual population data as illustrated by the graphics above (the model is descriptive [\neq mechanistic], but accurately reflects population heterogeneity and is easy to solve numerically by taking a matrix exponential of matrix A, respectively B). Thank you for pointing out the typo in Eq(2) and the discrepancy with Fig. 1! We corrected the typo in the equation and edited the figure accordingly. x_{19} is yellow to show that it is not infectious anymore but the virus is still detectable (by PCR). (line 110-158)

Population model

As matrix A, Matrix B is also unclear.

We expanded the matrix descriptions, corrected a typo in the matrices and changed the color and font of the helper text outside the matrices. Essentially, the equation in line 132 is the so-called ‘master equation’ for the within-host model in matrix form and A is the transition rate matrix (transpose of generator matrix). Matrix B adds an extra-state to matrix A (diagnosis state). This makes it possible to compute the time-dependent diagnosis probability alongside solving the within-host model.

“After an individual recovers, a new susceptible is introduced in the population pool, keeping the sum of infected + susceptible constant during a simulation (reinfection).”

Does this imply that after recovery, the individual quickly becomes entirely susceptible. This is surely not true for SARS-CoV-2 infection. Duration of immunity needs to be included.

We agree that for SARS-CoV-2 infection assuming individuals become immediately susceptible after infection is not realistic, as immunity against the variant causing infection is well documented, as well as cross immunity to other lineages. There are two reasons why we re-introduce ‘susceptibles’ into the population: (i) Because of computational restrictions, we

were limited with regards to the number of agents we could simulate. If we would not add susceptibles, they may quickly become entirely depleted and consequently the pandemic would go extinct. (ii) Escape variants enable re-infection of recovered individuals (e.g. PMID: 39880955). By adding back susceptibles and considering transmission advantages between strains with regards to infecting these susceptibles we keep the pandemic ongoing in simulations and, at the same time, can observe evolution. We now acknowledge this in the discussion (line 603-609).

Line: 152: what are “infectious agents”? please specify “positions in the genome”?

Changed “infectious agents” to lineages “within infectious individuals”. Rephrased for better clarity (line 178 - 190)

Line 153: define “nucleotide substitution”

Added to text (line 189).

Lines 156-7: “e is the nucleotide substitution rate” Better to avoid using e as any parameter because e is usually assigned for the natural constant.

True, thank you. Updated to only use NSR.

Line 165: Briefly introduce Jukes-Cantor model.

Added a visual explainer (line 203).

Equation (3): “each infected individual” Is x_0 infected? Fig 1 shows that x_0 is in white, --susceptible?

x_0 is infected but not infectious. We wanted to show this difference in the figure. We changed the group x_0 - x_4 to a different color to avoid confusion. See also graphic above with regards to the intra-host model

Line 175: what does “the reaction fires” imply?

Corrected to “...reaction happens...” (line 218).

Lines 178-181: “We assume that the rate of intra-host diversification is constant and not depending on the number of lineages already present inside the host. Each individual has a maximum number of lineages that they can host”

“not depending on” and “has a maximum number” cannot be true at the same time.

That is true, thanks for pointing it out. We updated and corrected the paragraph (line 208-237).

Line 182: why 5?

In SARS-CoV-2 infections the establishment of co-occurring intra-host lineages has been observed, in some cases due to different ecological niche colonization (e.g. different organs / tissues). We assume that the number of maximum co-occurring lineages is limited and varies across hosts and we wanted to include it in the model parametrization. In the work presented here we used a uniform distribution to assign the maximum lineage hosting capacity of individuals so as not to impose any hard assumption on the underlying biology of the process. We selected 5 as an upper bound that has not actually been observed in real life to keep it fairly unconstrained. In figure S1 one can see the effect of k_v on actual intra-host lineages numbers. In any case, in the results presented in this paper, we set the virus duplication rate to 0, as the intra host lineage emergence is outside the scope of the presented work, meaning the value of this parameter is not affecting the presented results. We nevertheless include this mechanism in SIMPLICITY as it is relevant in other possible scenarios. We modified and corrected the paragraph in the paper to make this clearer (line 208).

Line 187: how to model “within-host competition”?

Modeling detailed intra-host competition and evolution is outside the scope of the current work. Our modeling choice is based on the assumption that for acute respiratory viruses like SARS-CoV-2, the selective pressures at the intra-host and inter-host levels can be considered independent processes (<https://doi.org/10.1093/molbev/msad204>). In particular, Ruan et al. provide evidence that within-host and between-host selective pressures act independently and possibly antagonistically in SARS-CoV-2. Following the reviewers' suggestions, we have expanded the discussion in the manuscript. We now acknowledge this limitation, clarifying that selection is considered at the host population scale rather than the intra-host scale (line 625-631)

Lines 190-192: The authors seem to argue this in two mutual ways. which way is used or workable? What I mean in the sentence is:

one way: “an infecting virus variant is chosen in proportion to its relative transmission fitness” (lines 191-192); the other way: equation (4) implies that fitness depends on “the proportion of the population that had been infected”

The distinction is not clearly explained.

The two ways do not contradict each other: the relative transmission fitness is what determines the chance of transmission and its value depends on the state of the population. When the infection begins, the population is immunologically “naive” and there is no genetic imprinting nor acquired immunity. As people get infected and the proportion of individuals that had contact with the virus increases, the fitness score term that has more impact on the final fitness value changes accordingly. We have clarified this in the text (line 238).

Line 195: “virus lineages are equally capable at infecting...”, why? does this assume that all the lineages are of the same transmission fitness? This is not in agreement with “This assumes that as the virus evolves, new lineages emerge that are better at infecting new individuals.” (Linear fitness model)

As noted above, the assumption that virus lineages are equally capable of infecting individuals applies only during the early, immune-naive phase of the epidemic, when imprinting and acquired immunity have not yet shaped population susceptibility. Once immunity begins to accumulate, this assumption no longer holds: new lineages may be successful depending on how they can navigate the emerging immune landscape. Thus, we distinguish between two epidemiological phases: an initial *immune-naive phase*, where all lineages have comparable transmission potential, and a subsequent *immune-adapted phase*, where viral evolution and immune escape drive differences in lineage fitness. We have clarified this in the text (line 238).

Line 199: the expression given is not “the proportion of the population that had been infected by time t ” The correct one should be $(D(t)+R(t)+I(t))/N(t)$ with $N(t)$ representing the total number of individuals in the population.

Indeed the expression is referring to the proportion of the population that exited the infected compartment at time t (as it is meant to be), corrected in text (line 249)

Equation for $d(I(t), c_0)$ needs to be explained.

Added a clarification after the equation is introduced (line 251)

Line 215: why six months?

Changed to “months”. It is around 6 months due to how the weight function is parametrized but this should indeed not be mentioned here as it is parametrization dependent (line 266).

Equation (5): what are the values of parameters k_e and k_a ?

Added in the parametrization section (line 349).

Lines 224-5: some details of “mechanism of interaction...” should be specified.

Line 275: “In this model, we **thus** introduce a mechanism of interaction between the emerging lineage...”

This sentence refers to what we explained above: the model introduced is the mechanism of interaction, which is explained in detail.

Line 233: what is SSA?

Added at the beginning of Methods section (Gillespie/SSA explanation) (line 99)

Line 237: what is “virtual reaction”?

Added in text (line 285).

In the table for “Algorithm 1...”, add a table to list the definitions of all variables used.

Added to supplementary

Line 254: “Lineages are defined by a unique set of substitutions.” What is a unique set?

Added a clearer explanation (line 305).

Line 290: what is transition matrix $P(t)$?

Added in text (line 206).

Lines 310-11: “fixing all parameters except the nucleotide substitution rates” The values of all other parameters should be given. If there are any uncertainties in other parameters, how to confirm the robustness of the conclusion?

We added the parameter table to the supplement (it was in the code script made available with the publication, now it is more explicit). We checked the stability of the results with sensitivity analyses across many parameters throughout the development of SIMPLICITY, but in this case it is not a problem as the user would fit the NSR to the desired OSR for any set of desired parameters.

Lines 322-4: “R effective was computed by calculating the ratio of births to deaths events over a sliding time windows of 21 days.” Does this calculation agree with equation (7)?

The population average R effective corresponds well with the R value set for the simulation (within 10% variation), showing the population model is robust and behaves as expected. Added to the discussion (line 535).

Referee #3: Infectious disease modeling

This manuscript presents a modeling framework for studying viral diversification and evolution on the population scale, incorporating precise modeling of individuals hosts and their infection trajectories and histories.

Overall, I would say this manuscript presents a multifaceted model in a way that it is clear how each piece works. I have some concerns about particular modeling assumptions and approaches, which I detail below. It's important to consider that ultimately some of these may be a matter of taste/what I find interesting compared to the authors. I found no technical errors in the actual materials, and found it quite readable.

My chief confusion centers on how evolution occurs in the within host model. The authors state that they "only consider substitution events in within-host lineages that are relevant to transmission" (l. 144). This is a fine assumption as other work has suggested that transmission bottlenecks are comparatively much more important for evolution in respiratory pathogens (Morris 2020 eLife is one example). It's clear to me how substitution events occur and how lineages emerge, and then lineages can be substituted overtime if an individual reaches their maximum number of lineages. What is unclear currently is how lineages are selected for replacement. Is it simply the case you replace the lineage occurring at the smallest frequency? Does this not simply mean the infecting strain will always be present at the highest frequencies in the host?

The reviewer is correct in the assumption that our motivation for lineage replacement within hosts is based on perceived lineage frequency dynamics and transmission bottlenecks. However, we did not explicitly model lineage frequencies within the host, but rather assumed that (i) substitution events coincided with lineage replacement and that (ii) lineage diversification coincided with compartmentalization, which may occur in case of prolonged infection.

The reviewer is also partly correct with the second question: **On a short time scale** (time of infection to transmission) the infecting strain will most likely be the transmitted strain in the model (which reflects our knowledge on the amount of evolution occurring in transmission pairs, PMID: 35853960, PMID: 35062291).

We updated the intra host diversification paragraph in the Methods section to clarify (line 208) and we extended the discussion (line 625-631) and added Morris 2020 eLife.

The relationship between empirical nucleotide substitution rate and observed substitution rate is not clear to me currently. As I understand it NSR is estimated empirically using sequences, then a simulation is performed, then they estimate the OSR from the simulated sequence data. Should the OSR not simply be an unbiased estimator of the NSR? The authors claim no (l. 379), but the intuition does not follow for me from their explanation. Perhaps a schematic figure or a more detailed explanation of how the model is tuned to get real world OSR would be helpful here.

The nucleotide substitution rate is fitted so that the user can set an observed substitution rate in a simulation with given parameters to be equal to what is observed empirically in the real world. The nucleotide substitution rate is the rate at which substitutions are incorporated in the genome of lineages within the hosts. The observed substitution rate is what we observe (using a simple phylogenomics estimator, TempEst) after a simulation scenario was run and is the analogue to what we infer from phylogenetic methods in real world data analysis. We set the NSR so that the OSR corresponds to real data. The relation between the NSR and the OSR is influenced by transmission dynamics, rate of diagnosis (and isolation, sequencing) and selective pressure. We show that a logarithmic or exponential curve best describes their relationship (see goodness of fit table added to Supplementary).

I understand details of the infection model are taken from previous work (Van Der Toorn 2021); however, I think a slightly longer treatment of some of these modeling decisions is still warranted. E.g. why 20 compartments, why does transmission begin in compartment 5 etc. Even just an explanation of the biological interpretation of these choices I think will help.

Thank you. To shorten the manuscript we initially kept it shorter but we now dedicate some more space to explain the intra-host model and clarify these points, also in response to reviewer #2. (line 110).

Is it the case in this model that upon receiving a positive diagnosis an individual always self-isolates for the remainder of the infection? I would like to see how modeling results do/don't depend on this assumption. This assumption is less accurate in 2025 than it was several years ago. Perhaps a nice extension of this model could be to look at how within host dynamics influence infection severity (and thus rate of diagnosis).

The reviewer understood correctly that upon positive diagnosis we assume self-isolation. In the work presented here we decided to focus on a scenario that resembled the early years of the pandemic, where self isolation was mandatory upon positive tests; nevertheless one can set the model parameters to investigate also different scenarios.

The within-host dynamics influence on disease severity is something that would be definitely interesting to include, which we are considering for future versions of SIMPLICITY.

I like the way the results are presented in Figure 3.

Thank you!

Why bother describing how you might use AI tools to estimate viral fitness? It seems as though you have shown that you can get at this concept more mechanistically with greater success.

Thank you for the feedback :) We removed the section on the AI tools, as the reviewer (and also reviewer #1) correctly point out that this is not the scope of the article.

Minor Comments

- l. 69: "Stochastic Simulation" should not be capitalized

Corrected - capitalized acronyms letters

- l. 286: "allow to reproduce also..." confusing word order

Corrected.

We thank both the editorial team and the reviewers for their constructive feedback. We have incorporated all required changes, documented editorial requests and uploaded the original tex files. Many thanks for improving the overall quality of the manuscript.

Editorial comments:

We therefore invite you to revise your paper one last time to address the remaining concerns of our reviewers:

-Please address the last comments from Reviewer #1.

- Done, see below. Manuscript with track changes uploaded (editorial and reviewer based revisions).

-Please make sure you include the clarifications in regards to equation 4 you mentioned in the rebuttal letter to Reviewer #2 (previous lines 190-195).

- We clarified this point by re-writing the paragraph preceding eq.(4) and correcting one term in the description of the “baseline model (linear)”, i.e. correcting ‘infecting new individuals’ to ‘re-infecting individuals’. Thanks for pointing this out.

At the same time we ask that you edit your manuscript to comply with our format requirements and to maximise the accessibility and therefore the impact of your work.

* Please see the attached document for editorial requests for the final version (.docx file). Please ensure a completed version of this file is uploaded as a Related Manuscript with your final submission.

- Uploaded.

* Please review our final submission file checklist to ensure all necessary files are present with your final submission and to avoid delays in accepting your manuscript. For your reference, a style and formatting guide is available here and includes all of our style requirements.

- Done.

REVIEWERS' COMMENTS:

Reviewer #1 (Remarks to the Author):

I'm grateful to the authors for their attention to the comments and suggestions! By and large, I believe the authors acknowledge the scope and limitations of the work more accurately. Three small suggestions come to mind:

1. In line 592, the number "105" comes from a formatting error from a previous comment of mine, which originally conveyed " 10^5 " using superscript notation, but was rendered on the PDF incorrectly.

- We apologize for the typo, which was corrected in the revised version

2. In line 597: This might be true about eSIM3's inability to model intrahost evolution, but I would argue it is also true about SIMPLICITY, as I argued in point #2 of the review. In this sense, bringing it up in this context might not be the differentiating factor that you are looking for.

- We agree with the reviewer and rephrased to “, while using discrete-time forward simulation (which may result in numerical errors)”

3. Regarding the last point on the review, which I will quote in full below—discussing this in the manuscript could be useful to illustrate the limitations and future directions!

"Figure 4: Comparing these entropy results from the simulated data to what is calculated from real-world observational data from the pandemic would be valuable.

This is definitely something we have considered but decided against as the evolutionary model is simplified compared to real data and we are not considering other aspects of SARS-CoV-2 evolution that definitely affected what happened in the real world during the pandemic. Once long-shedders are added in the simulations and a more realistic evolutionary model is used the comparison with real world entropy data would make more sense and provide a valuable insight. Additionally, we could not find any entropy related studies that would be a sensible comparison for our model."

- We added a sentence in line 584: “However, while comparing model-generated entropy predictions to real-world observational data would be valuable, the currently utilized evolutionary model may be too simplistic, and many factors including the impact of long-shedders on viral evolution remain to be included into SIMPLICITY.”